# Understanding Signal Propagation in GNNs Via Observables

## Abstract

Graph Neural Networks (GNNs) perform computations on graphs by routing the signal information between regions of the graph using a graph shift operator or a message passing scheme. Often, the propagation of the signal leads to a loss of information, where the signal tends to diffuse across the graph instead of being deliberately routed between regions of interest. Two notions that depict this phenomenon are oversmoothing and oversquashing. In this paper, we propose an alternative approach for modeling signal propagation, inspired by quantum mechanics, using the notion of observables. Specifically, we model the place in the graph where the signal lies, how much the signal is concentrated at this place, and how much of the signal is propagated towards a location of interest when applying a GNN. Using these new concepts, we prove that standard spectral GNNs have poor signal propagation capabilities. We then propose a new type of spectral GNN, termed Schrödinger GNN, which we show has a superior capacity to route the signal between graph regions.

## 1 Introduction

Graph Neural Networks (GNNs) (46; 25) have emerged as powerful tools, enabling breakthrough applications across diverse domains including molecular science, physics simulations, social network analysis, and recommendation systems. A GNN is a layered architecture that takes a graph with node features, often referred to as the signal, and returns some output, e.g., another signal on the same graph. The hidden states of the signal across the layers can be interpreted as a gradual flow or propagation of the node features, since the GNN computes the signal at the next layer using local operations on the previous layer.

Often, to solve a problem on graphs, the GNN should be able to direct the propagation of the signal from certain regions of the graph to others. For example, the function of an enzyme is often understood through the notion of allosteric regulation: activation in one site of the enzyme (the receptor) changes the dynamics of the molecule, leading to some change in another site, called the active site. To be able to predict such a behavior using a GNN, the GNN should be able to propagate the signal about the binding site, which captures structural properties of the receptor, to the distant active site.

However, one limitation of typical GNNs is that the signal gets diffused in all directions the more layers are used in the network, rather than being propagated, or routed, in a coherent way between regions in the graph. This limits the applicability of typical GNNs when a deliberate routing of the signal is required to solve the task. Two standard notions that are commonly regarded as quantifying this phenomenon are *oversmoothing* (33; 38; 58; 43; 9) and *oversquashing* (1; 51; 2)

However, the first notion, oversmoothing, which is often quantified via the Dirichlet energy (48; 45), describes how quickly the signal varies, or oscillates, across the whole graph, not how much the signal can be kept concentrated, or coherent, when propagating it from one region to another. The second phenomenon, oversquashing, describes the phenomenon where long range information is compressed through topological bottlenecks. Hence, analyses of oversquashing are typically based on various definitions quantifying bottlenecks, e.g., curvature (51), Cheeger number (7; 11), and effective resistance (2). Hence, such an approach focuses on structural properties of the graph, and do not typically explicitly study how coherent the signal stays when routing it between regions. For further details on oversmoothing and oversquashing see Appendix A.3.

**Our contribution.** We aim to directly study how coherent the signal stays when it is routed between regions of the graph. For this, we propose in the paper an alternative way to model and probe different aspects of the content of the signal and its flow. Specifically, we model (i) the location in the graph where the signal lies, (ii) how much the content of the signal is concentrated about this location, and, (iii) how much of the signal is propagated from one location of the graph to another when applying a GNN. Our Signal Routing Measure directly quantifies the ability to transport mass, addressing the core issue of oversquashing where information fails to propagate across bottlenecks. These three concepts are defined via the notion of observables and their mean and variance, similarly to the approach in quantum mechanics. Measuring signal content using observables was also done in the past in the context of signal processing (31; 29; 30; 18). We prove that standard spectral GNNs have poor signal propagation capabilities: they keep the location of the content of the signal unchanged, and only increase the spread of the signal about this location. Then, we propose a novel spectral GNN, called *Schrödinger GNN*, which has provably good signal flow properties. Namely, with *Schrödinger filters*, we can direct the propagation of the signal in any desired direction in the graph.

Schrödinger GNNs are based on two main components: a unitary graph shift operator (GSO), and complex modulated signals. The unitary GSO is analogous to the Schrödinger operator in classical quantum mechanics, and specifically, in the free particle dynamics. It assures that the content of the signal is transformed in a geometry preserving way, rather than being diffused. Moreover, Schrödinger GNNs consider some of the input feature channels as encoding an abstract notion of ambient location in the graph. We call these features *formal locations*. The rest of the feature channels are called *the signal*. The idea is to be able to shift the signal across the formal location, in any desired direction. For illustration, in a social network, we might want to shift the *income* signal along the *age* direction, to allow comparing salaries of different age groups. To quantify the propagation properties of signals, we consider an observable corresponding to each formal location feature, namely, an operator that measures the formal location of signals. Moreover, to guarantee that the formal location of signals shifts when applying GNNs, we form in the signal complex oscillations along the direction of each formal location. We show that this leads roughly to a constant speed of the formal location of signals when applying linear Schrödinger filters.

We empirically validate our theory on graph classification and regression benchmarks, where Schrödinger GNNs achieve comparable accuracy to state of the art GNNs.

## 2 MEASURING SIGNAL LOCALIZATION AND PROPAGATION

**General Notations.** For $N \in \mathbb{N}$ we denote $[N] = \{1. \ldots, N\}$. We use lowercase $a$, bold $\boldsymbol{a}$, and uppercase $\boldsymbol{A}$ for scalars, vectors, and matrices respectively. We also treat vectors $\mathbf{f} = (f_1, \ldots, f_n) \in \mathbb{C}^N$ as functions $f : [N] \to \mathbb{C}$, where $f(n) = f_n$. The identity matrix is denoted by $\boldsymbol{I}$. For a matrix $\boldsymbol{A}$, we denote by $\boldsymbol{A}_{n,:}$ and $\boldsymbol{A}_{:,k}$ its $n$-th row and $k$-th column respectively. For complex numbers, we denote complex conjugation by $\overline{z}$, real part by $\mathrm{Re}(z)$, and imaginary part by $\mathrm{Im}(z)$. A graph is $\mathcal{G} = (V, E)$ where the vertex set is $V = [N]$ and $E \subset [N]^2$. We denote by $\mathcal{N}(v)$ the neighborhood of vertex $v$. We consider only undirected graphs, and denote the adjacency matrix by $\boldsymbol{A} = (a_{n,m})_{n,m} \in \mathbb{R}^{N \times N}$. A graph-signal is a pair $(\mathcal{G}, f)$ where $f = (f_1, \ldots, f_K) : V \to \mathbb{C}^K$ is the signal. The signal can also be represented by a matrix $\boldsymbol{X} = (x_{n,k})_{n,k} \in \mathbb{C}^{N \times K}$ where $x_{n,k} = f_k(n)$. A graph shift operator (GSO), is any operator that encodes the graphs structure, e.g., the adjacency matrix or any graph Laplacian. We define the inner product of two single-channel signals $f, g \in \mathbb{C}^N$ by $\langle f, g \rangle = \sum_{v \in V} f(v)\overline{g(v)}$, and define norm by $\|f\|_2^2 = \langle f, f \rangle$. The operator norm is $\|\boldsymbol{A}\| = \sup_{\|x\|_2=1} \|\boldsymbol{A}x\|_2$. For a signal $f$, we denote by $\mathrm{diag}(f)$ the diagonal matrix with diagonal elements $\mathrm{diag}(f)_{n,n} = f_n$. The commutator of two matrices is $[\boldsymbol{X}, \boldsymbol{Y}] = \boldsymbol{XY} - \boldsymbol{YX}$.

**Observables and The Signal Routing Measure.** In a general Hilbert space $\mathcal{H}$ of signals, an *observable* is a self-adjoint operator $A$ in $\mathcal{H}$, i.e. $A^* = A$. By the spectral theorem, any self-adjoint operator in a finite dimensional spaces can be written as $A = \sum_j \lambda_j P_j$ where $\{\lambda_j\}_j$ are real eigenvalues and $\{P_j\}_j$ are the orthogonal eigenprojections. This decomposition motivates treating a self-adjoint operator as an *observable* of a *physical quantity*. Namely, we interpret the eigenvalues as values that the physical quantity can attain, and $P_j$ as projections upon spaces of signals that have $\lambda_j$ as the value of their physical quantity. For example, the diagonal operator $D : \mathbb{C}^N \to \mathbb{C}^N$ defined

by $(Dg)_j = jg_j$ can be thought of as a *location observable* on the line $[1, N]$. Here, the eigenvectors $e_j = (0, \ldots, 0, 1, 0, \ldots, 0)$ (with 1 only at the $j$-th entry) are thought of as pure states/signals with location exactly $\lambda_j = j$. Any signal $g \in \mathbb{C}^N$ is a linear combination of the *pure location states* $\{e_j\}_j$, i.e., $g = \sum_j g_j e_j$ with $g_j \in \mathbb{C}$. When the state $g$ is normalized to $\|g\|_2 = 1$, we can interpret $|g_j|^2$ as the weight, or probability, of $g$ being at location $j$. While $g$ does not have one exact location, we can define its *mean location* as $\mathcal{E}_D(g) = \sum_j |g_j|^2 j$, and its location variance as $\mathcal{V}_D(g) = \sum_j |g_j|^2 (j - \mathcal{E}_D(g))^2$. Using operator notations, these two quantities can be written as $\mathcal{E}_D(g) = \langle Dg, g \rangle$ and $\mathcal{V}_D(g) = \|(D - \mathcal{E}_D(g)I)g\|_2^2$, where $I$ is the identity operator in $\mathbb{C}^N$.

This discussion motivates the general construction of observables in quantum mechanics. For a self-adjoint operator $A$ and normalized state $g \in \mathcal{H}$, the *expected value* (or *mean*) of $A$ with respect to $g$ is defined to be $\mathcal{E}_A(g) := \langle Ag, g \rangle$. Note that when $\mathcal{H} = \mathbb{C}^N$, we have $\mathcal{E}_A(g) = \sum_i \lambda_i \langle P_i g, g \rangle$, which is interpreted just like the above example of location observable. The *variance* of $A$ with respect to $g$ is defined to be

$$\mathcal{V}_A(g) := \|(A - \mathcal{E}_A(g)I)g\|_2^2 = \langle (A - \mathcal{E}_A(g)I)^2 g, g \rangle = \mathcal{E}_{A^2}(g) - \mathcal{E}_A(g)^2.$$

In addition to the classical notions of mean and variance, we propose quantifying how well a signal is transmitted towards a target value of the physical quantity. Consider a scenario where we have an initial signal $g_0$, and we would like to transmit this signal to be concentrated about some value $r$ with respect to some observable $A$. For that, suppose that we operate on $g_0$, e.g., with a GNN, and transform it to $g_t$. The following definition quantifies how well $g_t$ achieves this goal.

**Definition 2.1** (Signal Routing Measure). *For an observable $A$, normalized initial signal $g_0$ and final signal $g_t$, and a target value $r \in \mathbb{R}$, the* signal routing measure *is defined to be*

$$\mathcal{P}_A(g_0, g_t, r) = \frac{\langle (A - Ir)^2 g_t, g_t \rangle}{\mathcal{V}_A(g_0)}. \tag{1}$$

In the setting of Definition 2.1, the observable $A$ models some physical quantity. The term $\langle (A - Ir)^2 g_t, g_t \rangle$ quantifies how much the values of the physical quantity of $g_t$ are concentrated about $r$, and the denominator normalizes this with respect to how well the physical quantity of the initial state $g_0$ is concentrated. It is easy to verify the identity

$$\mathcal{P}_A(g_0, g_t, r) = \frac{\mathcal{V}_A(g_t) + (r - \mathcal{E}_A(g_t))^2}{\mathcal{V}_A(g_0)}. \tag{2}$$

Hence, to minimize the routing measure, one should construct an operation that transforms $g_0$ to some $g_t$, keeping the variance of $g_t$ small (relatively to the variance of $g_0$), while making the expected value of $g_t$ as close as possible to $r$.

## 3 SIGNAL PROPAGATION IN SCHRÖDINGER GRAPH SIGNAL PROCESSING

Next, we introduce Schrödinger graph signal processing, and analyze signal propagation under it.

**Feature Location Observables.** Consider a graph-signal $(\mathcal{G}, q)$ with $q = (q_1, \ldots, q_M) : V \to \mathbb{C}^M$. We treat some of the feature channels of $q$ as the signal and some as some abstract notion of locations. Namely, for some $1 < J < M$ we call $g = (q_1, \ldots, q_J)$ the *signal*, and call $f = (q_{J+1}, \ldots, q_M)$ the *feature locations*. Denote $K = M - J$ and $f = (f_1, \ldots, f_K)$. As we show later, working with complex-valued signals is important for routing signals between graph regions. Hence, we consider $g : V \to \mathbb{C}^J$ with $\|g_j\|_2 = 1$, and consider real-valued feature locations $f : V \to \mathbb{R}^K$, which need not be normalized. Define the *feature location observables* $X_{f_k} = \text{diag}(f_k)$, for $k \in [K]$. By the fact that $f_k$ is real-valued, $X_{f_k}$ is self-adjoint. Now, $\mathcal{E}_{X_{f_k}}(g_j) = \sum_{n \in [N]} f_k(n) |g_j(n)|^2$ is interpreted as the $f_k$-value about which the energy of $g_j$ is centered, and $\mathcal{V}_{X_{f_k}}(g_j)$ is the spread of the energy of $g_j$ about this center.

**Partial Derivatives and The Second Order Feature Derivative GSO.** Our construction of Schrödinger signal processing is based on a special constructions of a GSO based on derivatives.

**Definition 3.1** ($f_k$-partial derivative). *Given a feature location $f_k : V \to \mathbb{R}$, we define the $f_k$-partial derivative $\nabla_{f_k} \in \mathbb{C}^{N \times N}$ by: for $n, m \in V$*

$$(\nabla_{f_k})_{n,m} = a_{n,m}(f_k(n) - f_k(m))$$

It is easy to see that $\nabla_{f_k}$ is skew-symmetric (i.e. $\nabla_{f_k}^* = -\nabla_{f_k}$), and hence $\nabla_{f_k}^2$ is self-adjoint.

**Definition 3.2** (Schrödinger Laplacian). *Given $K$ feature locations $f = (f_1, \ldots, f_K)$, the corresponding* Schrödinger Laplacian *is defined to be*

$$\Delta_f = - \sum_{k \in [K]} \nabla_{f_k}^2.$$

The Schrödinger Laplacian is self-adjoint as a sum of bounded self-adjoint operators. This makes the following operator unitary.

**Definition 3.3** (Schrödinger Operator). *Given feature locations $f : V \to \mathbb{R}^K$ and time $t \in \mathbb{R}$, the corresponding* Schrödinger Operator *is defined to be $\mathcal{S}[t, f] = e^{-it\Delta_f}$.*

As we define in Section 3, Schrödinger graph signal processing is based on filtering signals using Schrödinger operators as GSOs. In this paper, we develop the theory for Schrödinger operators based on Schrödinger Laplacians, as these special GSOs lead to theoretical guarantees. However, the Schrödinger signal processing methodology works also with Schrödinger operators based on general GSOs, like standard Graph Laplacians.

Let us draw an analogy to the classical theory. In the free particle Schrödinger equation, we consider the space $\mathbb{R}^3$ as the "graph," consider the coordinates $x, y, z$ as the locations, and $\partial_x, \partial_y, \partial_z$ as the partial derivatives. Here, $\Delta_{x,y,z} = -\partial_x^2 - \partial_y^2 - \partial_z^2$ is the classical Laplace operator. Given a wave function $g_0 : \mathbb{R}^3 \to \mathbb{C}$ representing a particle at time 0, $g_t = \mathcal{S}[t; x, y, z]g_0$ is the particle at time $t$. In our case, given a signal $g^{(0)}$ on the graph, thought of as the state at time 0, we denote $g^{(t)} = \mathcal{S}[t, f]g$, thought of as the signal at time $t$.

**Analyzing Signal Propagation via Splitting.** Note that typical signals are not localized about one feature location. For example, the grayscale signal of an image is typically supported across all $x, y$ locations. Hence, the expected location and location variance are not meaningful localization notions for such signals (see Figure 1 for illustration). Still, we can conceptually apply a localization analysis with observables as follows. We decompose the signal $g$ into a sum of chunks $g = g^1, \ldots, g^L$, e.g., by multiplying the signal by a window in the formal locations $g^l = w^l(f)g$, where $w^1, \ldots, w^L : \mathbb{R} \to \mathbb{R}$ form a partition of unity. Here we assume that each $w_j$ is well localized about one location value. Then, each chunk $g^l$ has a meaningful mean location, and we can track how Schrödinger operators propagate this location. Moreover, by tracking how much the Schrödinger operator increases the variance of the chunk, we interpret how much the content of the signal in this chunk is diffused, scatters, or dispersed, when propagating it. Note that this analysis makes sense by the linearity of the Schrödinger operator. Note as well that in our methodology, we do not decompose $g$ to chunks in practice, and this decomposition is just for conceptualizing the signal propagation. In appendix F.4 we use the splitting scheme to diagnose the signal propagation capabilities of Schrödeinger GNNs..

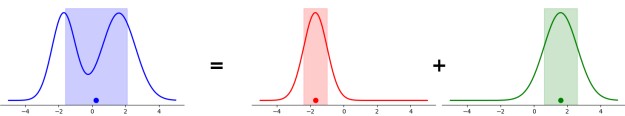

Figure 1: Decomposition of a signal $g$ to $g^0 + g^1$. Expected feature locations are marked by a dot, and the variance is signified by a color band.

**Dynamics of 1D Signals via Feature Momentum.** In the classical theory, the partial derivatives are called the *momentum observables*. The mean $i\mathcal{E}_{\partial_x}(g)$ is interpreted as the expected momentum, or speed, of the particle $g$. Analogously, we interpret the $f_k$-partial derivative $i\nabla_{f_k}$ as observables of

momentum or velocity along $f_k$. This interpretation can be made precise by developing dynamical equations of signals under Schrödinger operator, as we do next.

In the following discussion, we consider the case of single-channel signal $g = g_1$ and a single feature location $f = f_1$. We first show that the expected momentum of a signal is constant under Schrödinger dynamics.

**Theorem 3.4** (Constant Expected Momentum). *Let $g : V \to \mathbb{C}$ be a normalized signal and $f : V \to \mathbb{R}$ a feature location. Then, for every $t \in \mathbb{R}$,*

$$\mathcal{E}_{i\nabla_f}(g_t) = \mathcal{E}_{i\nabla_f}(g)$$

We then show that the rate of change of the expected location is equal to some smoothed version of the expected momentum. For that, we first define smoothing with respect to feature directions.

**Definition 3.5** ($f$-smoothing operator). *Let $f$ be a feature location. The $f$-smoothing operator $W_f : \mathbb{R}^N \to \mathbb{R}^N$ is defined as follows. For every signal $g \in \mathbb{C}^N$ and vertex $v \in V$*

$$(W_f g)(v) = \sum_{w \in \mathcal{N}(v)} a_{v,w}(f(w) - f(v))^2 g(w).$$

By definition, the $f$-smoothing operator mixes the values of the signal $g$ only along edges where the feature $f$ changes. It is hence interpreted as smoothing along the $f$ direction.

**Theorem 3.6** (Expected Feature Location Derivative under Schrödinger dynamics). *Let $g : V \to \mathbb{C}$ be a normalized signal and $f : V \to \mathbb{R}$ a feature location. Let $g^{(t)} = \mathcal{S}[t, f]g$ for every $t \in \mathbb{R}$. Then,*

$$\frac{\partial}{\partial t}\mathcal{E}_{X_f}(g^{(t)}) = -2\mathrm{Re}\big(\langle i\nabla_f g^{(t)}, W_f g^{(t)}\rangle\big). \tag{3}$$

The right-hand-side of (3) is interpreted as a smoothed version of the expected momentum $\mathcal{E}_{i\nabla_f}(g^{(t)}) = \langle i\nabla_f g^{(t)}, g^{(t)}\rangle$. Hence, Theorem 3.6 states that the rate of change of the expected location is equal to a smoothed expected momentum. In Appendix C.2, we show that for smooth enough signals, the rate of change of the expected location is close to the exact expected momentum. Since the expected momentum is constant, the theorem suggests that the rate of change of the expected location is roughly constant, as long as the signal stays smooth enough. This analysis hence justifies calling $i\nabla_f$ the momentum, or velocity, observable.

We note that Theorem 3.6 is analogous to the classical case, where the rate of change of the expected location of a free particle is equal to its expected momentum, which is constant. See Appendix B for more details.

**Achieving Translations via Feature Modulation.** We wish to be able to translate the expected feature location of signals using Schrödinger operators. In typical graph data, all features are real. However, as we show next, for real value signals, the expected momentum is always zero. Hence, given a real-valued signal, to be able to route it between feature regions, we must first modify it to be complex-valued. We do this via the feature modulation operator.

**Definition 3.7** (Feature Modulation). *Given a real-valued feature location $h : V \to \mathbb{R}$ and a phase $\theta \in \mathbb{R}$, the Feature Modulation Operator is defined to be $D[\theta h] = \mathrm{diag}(e^{i\theta h})$, where $e^{i\theta h}$ is the vector with entry $(e^{i\theta h})(v) = e^{i\theta h(v)}$ for node $v \in V$.*

Next, we show that modulating a real-valued signal gives nonzero expected momentum in general.

**Theorem 3.8** (Expected Momentum of Modulated Signal). *Given a signal $g : V \to \mathbb{R}$, feature locations $f, h : V \to \mathbb{R}$, and a phase $\theta \in \mathbb{R}$, the expected momentum of $D[\theta h]g$ satisfies*

$$\mathcal{E}_{i\nabla_f}(D[\theta h]g) = -2 \sum_{(m,n) \in E} a_{m,n}g(m)g(n)(f(n) - f(m))\sin(\theta(h(n) - h(m))). \tag{4}$$

Theorem 3.8 can be interpreted as follows. Consider the edge signals $e_{g,h}, e_f : E \to \mathbb{R}$ defined by

$$e_{g,h}(v, w) = g(v)g(w)\sin\big(\theta(h(w) - h(v))\big), \quad e_f(v, w) = f(v) - f(w).$$

The right-hand-side of (4) is the edge-space inner product $\langle e_{g,h}, e_f \rangle$. Hence, as long as we choose a modulating feature $h$ such that $e_{g,h}$ and $e_f$ are not orthogonal, the expected momentum of $D[\theta h]g$ will be nonzero.

**Dynamics of Multi-Channel Signals and Observables.**

**Theorem 3.9** (Expected multi-Feature Derivative). *Given the Schrödinger Laplacian $\Delta_f = -\sum_{k \in [K]} \nabla_{f_k}^2$ and a normalized signal g, we have*

$$\frac{\partial}{\partial t} \mathcal{E}_{X_{f_k}}(g^{(t)}) = -2\mathrm{Re}\big(\langle i\nabla_{f_k} g^{(t)}, W_{f_k} g^{(t)} \rangle\big) + \sum_{j \neq k} \left\langle [i\nabla_{f_j}^2, X_{f_k}] g^{(t)}, g^{(t)} \right\rangle. \tag{5}$$

Ideally, we would like the rate of change of the expected $X_{f_k}$ location to be a smoothed version of the expected $\nabla_{f_k}$ momentum. However, we see that in (5) there are additional cross terms. This leads to the following definition.

**Definition 3.10** ($\epsilon$-Commuting Features). *A sequence of feature locations $\{f_1, f_2, \ldots, f_K\}$ is said to be $\epsilon$-commuting if for every pair $i \neq j \in [K]$, the matrix $E_{i,j} = [X_{f_i}, \nabla_{f_j}] = X_{f_i}\nabla_{f_j} - \nabla_{f_j}X_{f_i}$ satisfies $\|E_{i,j}\|_{op} \leq \epsilon$.*

For a sequence of $\epsilon$ commuting features, the dynamics is

$$\left\| \frac{\partial}{\partial t} \mathcal{E}_{X_{f_k}}(g^{(t)}) - 2\mathrm{Re}\big(\langle i\nabla_f g^{(t)}, W_f g^{(t)} \rangle\big) \right\| \leq (K-1)\epsilon.$$

Hence, here as well we have the interpretation that for smooth enough signals, the rate of change of all expected locations are close to their corresponding expected momenta.

**Orthogonalizing The Feature Directions.** The signal $q : V \to \mathbb{R}^M$ in the raw data is not $\epsilon$-commuting in general. Hence, in Schrödinger GNNs, as a first step, we transform the feature $y$ to a sequence of features $f_1, \ldots, f_K$ which are $\epsilon$-commuting. For example, one can plug each node feature $q(n)$ into a simple MLP or a linear transformation $\Theta$, to obtain $f(n) = \Theta(q(n))$. The transformation $\Theta$ is optimized with respect to the following target.

**Definition 3.11** (Position-Momentum Optimization (PMO)). *Given a signal $q \in \mathbb{R}^{N \times M}$, a linear transformation $T \in \mathbb{R}^{M \times K}$, mapping $q$ to $f = (f_1, f_2, \ldots, f_K) = qT \in \mathbb{R}^{N \times K}$, is optimized w.r.t*

$$\min_{T \in \mathbb{R}^{M \times K}} \sum_{i \neq j}^{K} \|[\nabla_{f_j}^2, X_{f_i}]\|_{op}^2 + \lambda \sum_{k=1}^{K} \left( \|\nabla_{f_k}\|_\infty - 1 \right)^2,$$

*for some $\lambda > 0$.*

**Dynamics of the Variance.** Next, we derive the dynamics of the variance.

**Theorem 3.12** (Time Derivative of Variance). *Let $g : V \to \mathbb{C}$ be a signal and $f : V \to \mathbb{R}$ a feature location, and $\Delta_f = -\nabla_f^2$. The first-order derivative of variance with respect to time $t \in \mathbb{R}$ is*

$$\frac{\partial}{\partial t} \mathcal{V}_{X_f}(g^{(t)}) = \mathcal{E}_{i[\Delta_f, X_f^2]}(g^{(t)}) + 4\mathcal{E}_{X_f}(g^{(t)})\mathrm{Re}\big(\langle i\nabla_f g^{(t)}, W_f g^{(t)} \rangle\big)$$

This mirrors the classical Schrödinger equation dynamics where variance evolution depends on both the commutator $[\Delta, X^2]$ and the coupling between position and momentum. See Appendix B for the classical correspondence.

**Improving Signal Routing Through Modulation.** Here, we show that in typical situations modulating real-valued signals improve their signal routing measure. Consider the following setting. We have a multilayer network where at each layer $l$ we have a real-valued signal $g^{(l)}$ that we are allowed to modulate by choosing the free parameter $\theta_l \in \mathbb{R}$ in $D[\theta_l h]g^{(l)}$. We then propagate the signal via $\mathcal{S}[dt, f]D[\theta_l h]g^{(l)}$ for some small time step $dt$, and lastly apply a modulus nonlinearity to define the signal at the next layer $g^{(l+1)} = \left| \mathcal{S}[dt, f]D[\theta_l h]g^{(l)} \right|$. Here, we can interpret $g^{(l)}$ as the signal at time $ldt$, and the input to the network $g^{(0)}$ as the signal at time 0.

Suppose that we would like to rout the signal to the feature location $r$, i.e., we would like $\mathcal{P}_{X_f}(g^{(0)}, D[\theta_l h]g^{(l)}, r)$ to decrease in $l$ by choosing appropriate $\theta_l$. In this setting, since $dt$ is small, we can linearize the propagation of $g^{(l)}$ about $t = 0$, and obtain

$$\mathcal{P}_{X_f}(g^{(0)}, g^{(l+1)}, r) = \mathcal{P}_{X_f}(g^{(0)}, D[\theta_l h]g^{(l)}, r) + \frac{\partial}{\partial t}\mathcal{P}_{X_f}(g^{(0)}, \mathcal{S}[t, f]D[\theta_l h]g^{(l)}, r)|_{t=0}dt + O(dt^2)$$

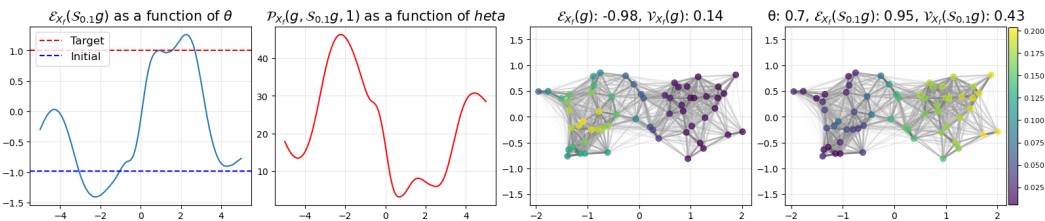

Figure 2: Signal transport under modulation.

$$= \mathcal{P}_{X_f}(g^{(0)}, g^{(l)}, r) + \frac{\partial}{\partial t}\mathcal{P}_{X_f}(g^{(0)}, \mathcal{S}[t, f]D[\theta_l h]g^{(l)}, r)|_{t=0}dt + O(dt^2),$$

where the last equality is due to the fact that $\mathcal{P}_{X_f}(g^{(0)}, D[\theta_l h]g^{(l)}, r)$ does not depend on $\theta_l$. We would now like to know if modulating the signal at layer $l$ improves the routing measure at layer $l+1$. For that, it is enough to show that the derivative of $\mathcal{P}_{X_f}(g^{(0)}, g^{(l+1)}, r)$ with respect to $\theta_l$ is nonzero at $\theta_l = 0$. Observe that

$$\frac{\partial}{\partial \theta_l}\mathcal{P}_{X_f}(g^{(0)}, g^{(l+1)}, r) = \frac{\partial}{\partial \theta_l}\frac{\partial}{\partial t}\mathcal{P}_{X_f}(g^{(0)}, \mathcal{S}[t, f]D[\theta_l h]g^{(l)}, r)|_{t=0} + O(dt^2).$$

Hence, our goal is to show that $\mathcal{D} := \frac{\partial}{\partial \theta_l}\frac{\partial}{\partial t}\mathcal{P}_{X_f}(g^{(0)}, \mathcal{S}[t, f]D[\theta_l h]g^{(l)}, r)|_{t,\theta_l=0}$ is nonzero in general. As long as this is true, $\theta_l = 0$ is not the minimizer of $\mathcal{P}_{X_f}(g^{(0)}, g^{(l+1)}, r)$, so one can always choose a better modulation than $\theta_l = 0$.

We now simplify the notations and give a formula for $\mathcal{D} := \frac{\partial}{\partial \theta}\frac{\partial}{\partial t}\mathcal{P}_{X_f}(g, \mathcal{S}[t, f]D[\theta h]g, r)|_{t=\theta=0}$.

**Claim 3.13** (Mixed Derivative of The Signal Routing Measure)**.**

$$\frac{\partial}{\partial \theta}\frac{\partial}{\partial t}\mathcal{P}_{X_f}(g, \mathcal{S}[t, f]D[\theta h]g, r)\Big|_{t=\theta=0} = \frac{\langle [X_h, [\Delta, X_f^2]]g, g\rangle \; + \; 4r\,\mathrm{Re}\,\langle [X_h, W_f\nabla_f]g, g\rangle}{\mathcal{V}_{X_f}(g)}$$

We see that when $h$ is constant, i.e. there is no modulation, there is no modulation, $\mathcal{D}$ is zero.

In Figure 2 we give an example of a graph, initial signal $g$ with $\mathcal{E}_{X_f}(g) = -0.98$, modulating feature $h = f$, and desired location value $r = 1$. We show that by choosing an appropriate modulation $\theta$ and propagating the signal using the Schrödinger operator to time $t = 0.1$ improves the signal routing measure with respect to not modulating.

**Schrödinger Signal Processing.** We define Schrödinger filters by considering linear combinations of the evolutions of the modulated signal with different modulations and times. Let $f : V \to \mathbb{R}^K$ be location features and $D \in \mathbb{N}$ be the output feature dimension. To use linear algebra notations, let us now treat signals and location features and vectors in $\mathbb{C}^{N \times J}$ and $\mathbb{R}^{N \times K}$ respectively. A Schrödinger filter $\Psi$ is parameterized by $(t_m \in \mathbb{R}, \theta_m \in \mathbb{R}, \mathbf{W}^{(m)} \in \mathbb{C}^{J \times D}, \mathbf{T}^{(m)} \in \mathbb{R}^{K \times 1})_{m \in [M]}$, and maps signals $\mathbf{g} \in \mathbb{C}^{N \times J}$ to

$$\Psi(\mathbf{g})\mathcal{F}(\mathbf{g}) = \sum_{m=1}^{M}\mathcal{S}[t_m, \mathbf{f}]D[\theta_m\mathbf{f} \cdot \mathbf{T}^{(m)}]\mathbf{g} \cdot \mathbf{W}^{(m)}.$$

**Schrödinger GNNs.** The application of a Schrödinger GNN is a two-step procedure. First, the input features are optimized via Position-Momentum Optimization (PMO) (Definition 3.11) to obtain the location features $f$. Second, the Schrödinger GNN is trained using these fixed features. For nonlinearities within the network, we apply standard activations (e.g., ReLU) separately to the real and imaginary parts: $\sigma(z) = \mathrm{ReLU}(\mathrm{Re}(z)) + i \cdot \mathrm{ReLU}(\mathrm{Im}(z))$ or we used the absolute value $\sigma(z) = |z|$. See Appendix E.3 for full implementation details and Appendix E.3 for computational complexity analysis.

**Uniform Time initialization.** Schrödinger layers include a per channel real scaling parameter $t \in \mathbb{R}^{C_{\text{out}}}$. At initialization we draw each channel independently $t_j \sim \mathrm{Uniform}(0, 1.5)$. Larger $t$

increases the contribution of higher order propagation steps (capturing longer range interactions), whereas smaller $t$ biases updates toward local mixing. When learning is disabled we use a non-trainable scalar 1.0. We coined the name Adaptive Unitary for the layer with only the Unitary Schrödinger operator with different learnable $t_j$ without modulation layer, In depth explanation in the Appendix E.3.

## 4 EXPERIMENTS

**Synthetic Experiment - Signal Propagation on a Cycle.** Here, we showcase the capability of Schrödinger GNN to direct the propagation of the signal with a toy regression experiment. Consider a cycle graph discretizing the unit circle, and the locations feature $x = \cos(\theta)$, where $\theta$ is the angle. Each signal in the dataset is a Gaussian with random mean $\mu$ and variance $\sigma^2$, and with additive white noise. The target for each signal is the same gaussian mean shifted by a predetermine value $d$. The task is to learn a GNN that maps the input signal to the output sig-

Table 1: Test Losses for Ring Signal Transport

| Model | Test Loss |
|---|---|
| GCN (25) | $0.6644 \pm 0.0720$ |
| GAT(53) | $0.6050 \pm 0.0052$ |
| Schrödinger real | $0.9334 \pm 0.0514$ |
| Schrödinger | **3e-04 $\pm$ 2e-04** |

nal. This experiment shows that only Schrödinger GNN, with modulated input signal, can solve this task. A summary of dataset statistics is available in Appendix F.3.

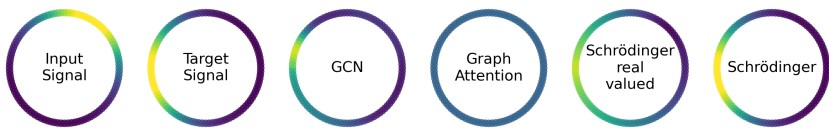

Figure 3: Cycle graph (ring) signal transport. Each panel is a cycle graph in which node color intensity encodes the signal magnitude. All panels share the same color scale.

**MNIST Classification** We conduct an experiment on the classical MNIST dataset (28) to evaluate our model's performance on a standard image classification task formulated as a graph problem. Each image is converted into a graph where each pixel is a node. Node features include the pixel's intensity and its (x, y) coordinates. Edges connect each pixel to its eight closest neighbors. We ran each model five times for 200 epochs. As shown in Table 2, our Schrödinger model achieves competitive performance. Further details are provided in Appendix F.4.

Table 2: MNIST classification results (Test Accuracy). Results averaged over 5 runs.

| MODEL | ACCURACY |
|---|---|
| GCN (25) | $92.09 \pm 0.28$ |
| ChebConv (12) | $95.72 \pm 0.74$ |
| GAT (53) | $95.94 \pm 0.71$ |
| GIN (56) | $98.33 \pm 0.11$ |
| MPNN (16) | $98.95 \pm 0.06$ |
| CNN (27) | $99.07 \pm 0.07$ |
| Schrödinger | **99.13 $\pm$ 0.04** |

**Graph Classification - Architecture Matched Comparison** To ensure a fair comparison across different GNN architectures, we conduct an additional evaluation on ENZYMES, IMDB, MUTAG, and PROTEINS using a standardized architecture: three inner convolution layers followed by a final linear layer. For fairness, we match the parameter count across all methods by first computing the parameter count of a GCN model with hidden dimension 128, then adjusting the hidden dimensions of all other methods GAT, Unitary, Adaptive Unitary, Schrödinger, Schrödinger PMO (Position-Momentum Optimization before training) to match this parameter count within 0.6% tolerance. This ensures that performance differences reflect architectural choices rather than model capacity. Each model-dataset combination was run 100 times with different random seeds, and the reported results show the mean and standard deviation across these runs. Results are reported in Table 3, for more details F.4.

**Peptides** Peptide-Func and Peptide-struct, two datasets taken from Long Range Graph Benchmark (LRGB) (14) comprise datasets that specifically test the ability of graph neural networks to capture long-distance dependencies between nodes. For this paper, we focus on the molecular property

Table 3: Architecture-matched comparison results (Test AP ↑). All models use 3 convolution layers + 1 linear layer with matched parameter counts. Top-1/2/3 entries are highlighted green/orange/yellow, respectively.

| Model | ENZYMES | IMDB | MUTAG | PROTEINS |
|---|---|---|---|---|
| GIN (56) | $31.93 \pm 3.16$ | $69.22 \pm 3.14$ | $78.19 \pm 5.57$ | $71.88 \pm 3.08$ |
| GCN (25) | $31.66 \pm 5.35$ | $50.6 \pm 4.1$ | $73.24 \pm 6.27$ | $71.41 \pm 3.04$ |
| GAT (53) | $31.13 \pm 34.88$ | $49.54 \pm 2.54$ | $75.21 \pm 6.41$ | $72.31 \pm 3.28$ |
| Unitary (UniGCN) (23) | $40.3 \pm 6.63$ | $65.42 \pm 2.8$ | $75.74 \pm 6.67$ | $69.19 \pm 3.01$ |
| Adaptive Unitary | $41.6 \pm 5.18$ | $65.46 \pm 2.48$ | $75.53 \pm 5.95$ | $71.79 \pm 3.33$ |
| Adaptive Unitary PMO | $41.83 \pm 4.44$ | $66.27 \pm 3.01$ | $75.62 \pm 6.24$ | $7.177 \pm 2.84$ |
| Schrödinger | $43.5 \pm 4.89$ | $65.86 \pm 2.83$ | $75.42 \pm 6.11$ | $71.57 \pm 2.56$ |
| Schrödinger PMO | $43.7 \pm 3.37$ | $69.6 \pm 2.85$ | $79.25 \pm 6.19$ | $72.68 \pm 3.05$ |

prediction datasets Peptides-func and Peptides-struct. Peptides-func is a graph-level classification task that determines functional characteristics of peptide molecules represented as graphs, while Peptides-struct is a graph-level regression task that predicts structural properties of these molecules, for more details F.8.

Table 4: Performance on Peptides-Func and Peptides-Struct. **Bold** values indicate the best performing models for each metric: the highest AP for Peptides-Func and the lowest MAE for Peptides-Struct. Top-1/2/3 entries are highlighted green/orange/yellow, respectively. The results for the models other than ours were taken from (20).

| MODEL TYPE | MODEL | PEPTIDES-FUNC (AP ↑) | PEPTIDES-STRUCT (MAE ↓) |
|---|---|---|---|
| MP | GCN[†] (25) | $68.60 \pm 0.50$ | $0.2460 \pm 0.0007$ |
| | GINE[†] (56) | $66.21 \pm 0.67$ | $0.2473 \pm 0.0017$ |
| | GatedGCN[†] (3) | $67.65 \pm 0.47$ | $0.2477 \pm 0.0009$ |
| | GUMP[‡] (41) | $68.43 \pm 0.37$ | $0.2564 \pm 0.0023$ |
| Others | GPS[†] (42) | $65.34 \pm 0.91$ | $0.2509 \pm 0.0014$ |
| | DRew[‡] (17) | $71.50 \pm 0.44$ | $0.2536 \pm 0.0015$ |
| | Exphormer[‡] (47) | $65.27 \pm 0.43$ | $0.2481 \pm 0.0007$ |
| | GRIT[‡] (35) | $69.88 \pm 0.82$ | $0.2460 \pm 0.0012$ |
| | Graph ViT[‡] (22) | $69.42 \pm 0.75$ | $0.2449 \pm 0.0016$ |
| | CRAWL[‡] (34) | $70.74 \pm 0.32$ | $0.2506 \pm 0.0022$ |
| | UniGCN[‡] (23) | $70.72 \pm 0.0035$ | $0.2425 \pm 0.0009$ |
| | Lie UniGCN[‡] (23) | $71.73 \pm 0.0061$ | $0.2460 \pm 0.0011$ |
| Ours | Schrödinger | $72.07 \pm 0.0099$ | $0.2439 \pm 0.00122$ |
| | Adaptive Unitary | $71.29 \pm 0.527$ | $0.2467 \pm 0.0011$ |

[†]Reported performance taken from (52). [‡]Reported performance taken from (23).

# 5 SUMMARY

We presented a new approach for defining and analyzing signal propagation across graphs. The approach directly models where the information of the signal is, how well concentrated it is, and how well it is routed between regions in the graph. We presented Scrödinger GNN, a graph neural network that is able to route the information of the signal along any direction in the graph. We showed that standard GNNs do not have this capability. One limitation of Scrödinger filters with respect to simple polynomial filters is that applying the Scrödinger operator on a signal involves approximating the exponential of the GSO, which involves applying the GSO several times.

# 6 ETHICS STATEMENT

This work presents theoretical and empirical contributions to graph neural networks using quantum-inspired methods. All experiments use synthetic data or publicly available benchmarks (LRGB Peptides, node classification datasets) with no privacy concerns or potential harm to subjects. The research involves only technical graph data and raises no ethical concerns.

# 7 REPRODUCIBILITY STATEMENT

We provide detailed proofs for all theorems, with additional analysis in the appendices. Implementation details including matrix exponential computation (Appendix E.1), hyperparameters (Appendix F.8), and synthetic experiment setups (Appendices F.3, F.1) are fully documented. The Position-Momentum Optimization is specified in Definition 3.11. Source code will be released on GitHub upon publication.

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

## A  BACKGROUND AND RELATED WORK

### A.1  SPECTRAL GNNS

Spectral GNNs define graph convolutions via the spectral domain. Let $\Delta$ be a self-adjoint GSO with $\{v_j\}_{j=1}^N$ and $\{\lambda_j\}^N$ which are the eigenvectors and eigenvalues so that $\Delta = \sum_{i=1}^N \lambda_i \, \mathbf{v}_i \mathbf{v}_i^\top$. Given a signal $\boldsymbol{X} \in \mathbb{R}^{N \times d}$ and a function $\boldsymbol{Q} : \mathbb{R} \to \mathbb{R}^{d' \times d}$, the spectral filter $\boldsymbol{Q}(\Delta) : \mathbb{R}^{N \times d} \to \mathbb{R}^{N \times d'}$ is defined by

$$\boldsymbol{Q}(\Delta)\,\boldsymbol{X} \ := \ \sum_{i=1}^N \mathbf{v}_i \mathbf{v}_i^\top \, \boldsymbol{X} \, \boldsymbol{Q}(\lambda_i)^\top. \tag{6}$$

A spectral GNN layer then applies $\boldsymbol{X}^{\ell+1} = \sigma\big(\boldsymbol{Q}_\ell(\Delta)\,\boldsymbol{X}^\ell\big)$ with trainable $\boldsymbol{Q}_\ell$ and nonlinearity $\sigma$. For more examples (4; 12; 32).

### A.2  UNITARY GNNS

Unitary GNNs are a class of graph neural networks designed to address fundamental challenges in deep graph learning, particularly oversmoothing and oversquashing, through the use of unitary transformations that preserve signal norms and maintain feature distinctiveness across layers. Known methods include Graph Unitary Message Passing (GUMP) (41) which transforms the adjacency matrix to be unitary, Unitary Group Convolutions (UGConvs) (59) which apply unitary transforms on groups, and Separable Unitary Convolution (UniConv/UniGCN) (23) which employs a unitary graph convolution. While UniConv utilizes a parameterization of unitary matrices (often based on Cayley transforms or Lie algebra generators) to maintain norm preservation, it fundamentally acts as a mixing operation within the spectral domain. In contrast, our Schrödinger GNN leverages the unitary operator specifically as a time evolution operator generated by a feature dependent Hamiltonian. This allows for directional signal routing steered by the underlying potential (the features), rather

than just mixing. Furthermore, Schrödinger GNN separates the "location" and "signal" aspects, optimizing the location features to maximize transport capability, a mechanism absent in standard unitary GNNs.

### A.3 OVERSMOOTHING AND OVERSQUASHING

Most works addressing the over-smoothing and over-squashing problems begin by considering the basic architecture of graph neural networks, the Message Passing Neural Network (MPNN) (16).

**Definition A.1** (Message Passing Neural Network). *Given a graph $G = (V, E)$ with node features $X \in \mathbb{R}^{N \times d}$, an MPNN updates node representations through:*

$$h_v^{(\ell+1)} = \phi_\ell \left( h_v^{(\ell)}, \sum_{w \in \mathcal{N}(v)} \psi_\ell(h_v^{(\ell)}, h_w^{(\ell)}) \right)$$

*where $h_v^{(0)} = x_v$, $\phi_\ell$ is the update function, and $\psi_\ell$ is the message function.*

Over-smoothing in GNNs refers to the tendency of node representations to become indistinguishable as network depth increases (44). The Dirichlet energy provides a standard measure for this phenomenon

**Definition A.2** (Dirichlet Energy). *For a signal $f : V \to \mathbb{R}$ and normalized Laplacian $\tilde{\Delta}$, the Dirichlet energy is*

$$\langle f, \tilde{\Delta} f \rangle = \frac{1}{2} \sum_{(i,j) \in E} w_{ij} \left( \frac{f(i)}{\sqrt{d_i}} - \frac{f(j)}{\sqrt{d_j}} \right)^2$$

*where $w_{ij}$ are edge weights and $d_i$ is the degree of vertex $i$.*

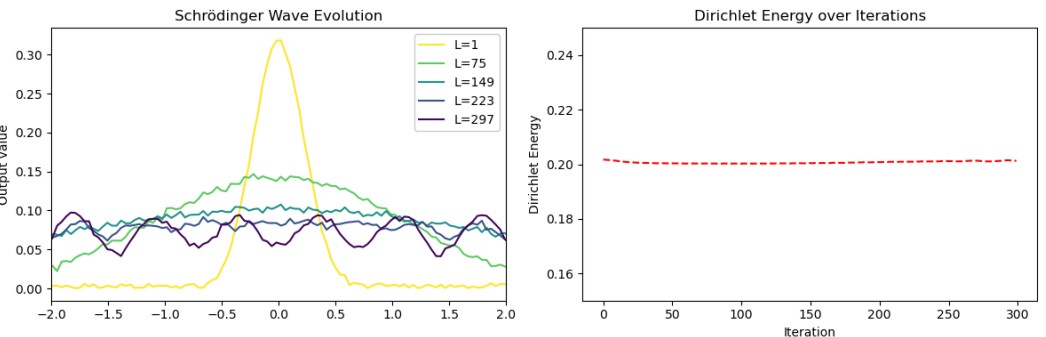

Figure 4: Evolution of a Gaussian signal on a ring graph under a unitary operator. The left plot shows the signal at different iterations (L), demonstrating that the signal's structure is preserved and does not smooth out. The right plot shows that the Dirichlet energy remains constant throughout the evolution. While unitary operators preserve Dirichlet energy, this example illustrates that it is more accurately described as a measure of oscillation rather than a measure of oversmoothing, as the signal maintains its local structure.

While Dirichlet energy has emerged as the dominant measure for analyzing over-smoothing in GNNs (44), it provides only a partial view of signal propagation dynamics. Dirichlet energy was first introduced to the GNN literature as a measure of signal smoothness across graph structures (5). It has since become the standard tool for analyzing over-smoothing phenomena. In the context of quantum mechanical observables, Dirichlet energy can be interpreted as the expected value of the observable Laplacian operator. However, this observable fundamentally measures the rate of change between neighboring nodes, essentially capturing local gradient information in the spatial domain, which corresponds to momentum space properties (see Theorem G.1). This perspective reveals critical limitations of Dirichlet energy: its local focus only captures immediate neighborhood relationships, missing long-range dependencies crucial for understanding over-squashing phenomena and signals whose mass is concentrated in specific graph neighborhoods. For GNN analysis, it is

beneficial to have the ability to quantify signal "transport" or understand relative signal localization.

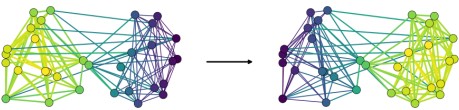

Figure 5: signal transport

Beyond the well known *over-smoothing* effect, MPNNs also suffer from *over-squashing*, where long-range information is compressed through topological bottlenecks and becomes effectively invisible to distant nodes. (1) showed first heuristics of over squashing and claim that the cause of bottlenecks is due the exponential growth of the node receptive field (8)

**Definition A.3** (Node Receptive Field Set). *Given graph $\mathcal{G} = (V, E)$, $r \in \mathbb{N}$ and node $v \in V$ the Receptive Field is*

$$B_r(v) := \{w \in V : d_G(v, w) \leq r\},$$

*where $d_G$ is the shortest path length on the graph*

(1) argued that oversquashing occurs when exponentially many messages are compressed into fixed-size vectors. (51) formalized this via sensitivity analysis:

**Definition A.4** (Oversquashing via Sensitivity). *Oversquashing occurs when the representation $h_v^{(\ell)}$ at node $v$ fails to be sufficiently affected by input features $x_w$ of distant nodes $w$. This is measured by the Jacobian $\|\partial h_v^{(\ell)} / \partial x_w\|$.*

**Lemma A.5** (r- distance Sensitivity Bound (51)). *Let $S_r(v) := \{w \in V : d_G(v, w) = r\}$. For an MPNN with bounded gradients $\|\nabla \phi_\ell\| \leq \alpha$ and $\|\nabla \psi_\ell\| \leq \beta$, if $w \in S_{r+1}(v)$, then*

$$\left\| \frac{\partial h_v^{(r+1)}}{\partial x_w} \right\| \leq (\alpha\beta)^{r+1} (A^{r+1})_{vw} \tag{7}$$

*where $A$ is the adjacency matrix and $(A^{r+1})_{vw}$ counts paths of length $r+1$ from $w$ to $v$.*

This bound reveals oversquashing, when $(A^r)_{vw}$ decays exponentially with distance (e.g., in trees), distant nodes have vanishing influence, creating information bottlenecks. (51) also connects to the Cheeger constant,

$$2h_G \geq \lambda_1 \geq \frac{h_G^2}{2}$$

which is a result from the Cheeger constant (7; 11)

$$h_G := \min_{S \subset V} h_S, \quad h_S := \frac{|\partial S|}{\min\{\text{vol}(S), \text{vol}(V \setminus S)\}}$$

and to the Cheeger inequality,

$$2h_G \geq \lambda_1 \geq \frac{h_G^2}{2}$$

which bounds the spectral gap. Here, $\lambda_1$ is the first non-zero eigenvalue of the normalized Laplacian; $\partial S = \{(i, j) : i \in S, j \in V \setminus S\}$; and $\text{vol}(S) = \sum_{i \in S} d_i$. The spectral gap can be interpreted as how well two partitions of a graph are connected. They use the spectral gap to support their graph curvature method and argue that negative edge curvature indicates its potential role in contributing to the oversquashing issue.

$$\text{Ric}(i, j) = \frac{2}{d_i} + \frac{2}{d_j} - 2 + 2 \frac{|\#_\triangle(i, j)|}{\max\{d_i, d_j\}}$$

where $\#_\triangle(i,j)$ counts triangles containing edge $(i,j)$. Negative curvature indicates potential over-squashing bottlenecks. Later work argued that not only edges are an indicator of oversquashing, but the relation between every two nodes on the graph. (2) base their method also on the spectral gap, and showcase their form of measure between two nodes, the effective resistance

**Definition A.6** (Effective Resistance). *For two nodes $u, v \in V$ their effective resistance is*

$$R_{u,v} = (1_u - 1_v)^\top \Delta^\dagger (1_v - 1_u)$$

where $\Delta^\dagger$ is the pseudoinverse of the graph Laplacian.

(2) generalized the sensitivity analysis to arbitrary node pairs using effective resistance:

**Lemma A.7** (Effective Resistance Sensitivity Bound). *For an MPNN with bounded gradients* $\|\nabla\phi_\ell\| \leq \alpha$ *and* $\|\nabla\psi_\ell\| \leq \beta$, *the sensitivity between nodes* $u, v$ *at layer* $r$ *satisfies:*

$$\left\| \frac{\partial h_v^{(r)}}{\partial x_u} \right\| \leq (\alpha\beta)^r \cdot exp\left(-c \cdot r \cdot R_{u,v}\right)$$

*where $R_{u,v}$ is the effective resistance and $c > 0$ is a constant depending on the graph.*

This bound shows that sensitivity decays exponentially with both distance and effective resistance, providing a more refined measure than path counting alone.

While these methods analyze oversquashing from graph topology, we propose that the choice of graph shift operator (GSO) also critically affects susceptibility to oversquashing. Different GSOs encode distinct notions of signal propagation, making some inherently more prone to information bottlenecks than others.

# B  SCHRÖDINGER IN CLASSICAL QUANTUM MECHANICS

Our graph based Schrödinger framework extends classical quantum mechanics. Understanding the classical case provides intuition for why real-valued graph signals require modulation to achieve directional transport, and establishes the theoretical foundations for our propagation measures. In this section, we establish the classical quantum mechanical foundations using our graph notation for consistency. Here, $g$ represents a continuous wavefunction $g : \mathbb{R} \to \mathbb{C}$, the feature location $f(x) = x$ is the spatial coordinate, and $X_f$ is the position operator acting as $(X_f g)(x) = x \cdot g(x)$. This can be understood both mathematically and intuitively: a real wave function represents a standing wave with equal probability of movement in opposite directions, resulting in no net momentum. More formally, for a real-valued wave function $g(x)$, we have

$$\mathcal{E}_{i\nabla_f}(g) = \langle g, i\nabla_f g \rangle = -i\hbar \int g(x) \frac{\partial}{\partial x} g(x) dx = 0$$

This property presents a challenge when we want to model directional information flow in graph neural networks, as real-valued node features would similarly lack directional momentum. We wish to understand how the wave function evolves in the classical case, so we need to understand the expected location derivative, also known as the Heisenberg motion equation.

**Theorem B.1** (Heisenberg Equation of Motion for Expected Values). *Let $g_t = \mathcal{S}_t g$ where $\mathcal{S}_t = e^{-it\Delta}$ is the Schrödinger evolution operator with Hamiltonian $\Delta$. For any observable $A$, the derivative of its expected value with respect to $t$ is*

$$\frac{\partial}{\partial t}\mathcal{E}_A(g_t) = i\langle [\Delta, A]g_t, g_t \rangle$$

*Proof.* We prove this using the limit definition and the expansion of the Schrödinger operator

$$\frac{\partial}{\partial t}\mathcal{E}_A(g_t) = \lim_{h \to 0} \frac{\mathcal{E}_A(g_{t+h}) - \mathcal{E}_A(g_t)}{h}$$

Since $g_{t+h} = \mathcal{S}_h g_t$ and $\mathcal{S}_h = e^{-ih\Delta}$:

$$\mathcal{E}_A(g_{t+h}) = \langle Ag_{t+h}, g_{t+h} \rangle = \langle A\mathcal{S}_h g_t, \mathcal{S}_h g_t \rangle = \langle \mathcal{S}_{-h} A\mathcal{S}_h g_t, g_t \rangle$$

Expanding $\mathcal{S}_h = e^{-ih\Delta} = I - ih\Delta + O(h^2)$ and $\mathcal{S}_{-h} = I + ih\Delta + O(h^2)$:

$$\mathcal{S}_{-h}A\mathcal{S}_h = (I + ih\Delta)A(I - ih\Delta) + O(h^2)$$
$$= A + ih\Delta A - ihA\Delta + O(h^2)$$
$$= A + ih[\Delta, A] + O(h^2)$$

Taking the limit

$$\frac{\partial}{\partial t}\mathcal{E}_A(g_t) = \lim_{h\to 0}\frac{\langle(A + ih[\Delta, A])g_t, g_t\rangle - \langle Ag_t, g_t\rangle}{h} = i\langle[\Delta, A]g_t, g_t\rangle$$

$\square$

**Theorem B.2** (Expected Position Evolution in Classical Case). *Let $g_t = \mathcal{S}_t g$ with $\mathcal{S}_t = e^{-it\Delta}$ where $\Delta = -\frac{\partial^2}{\partial x^2}$. Then the expected position evolves linearly with $t$*

$$\mathcal{E}_{X_f}(g_t) = \mathcal{E}_{X_f}(g_0) - 2t\mathcal{E}_{i\nabla_f}(g_0)$$

*Proof.* From Theorem B.1, we have

$$\frac{\partial}{\partial t}\mathcal{E}_{X_f}(g_t) = i\langle[\Delta, X_f]g_t, g_t\rangle$$

Computing the commutator $[\Delta, X_f] = [-\frac{\partial^2}{\partial x^2}, X_f]$: for any function $h$,

$$[-\frac{\partial^2}{\partial x^2}, X_f]h = -\frac{\partial^2}{\partial x^2}(xh) + x\frac{\partial^2 h}{\partial x^2} = -2\frac{\partial h}{\partial x} = 2i(i\frac{\partial h}{\partial x}) = 2i(i\nabla_f h)$$

Therefore $[\Delta, X_f] = 2i(i\nabla_f)$ and

$$\frac{\partial}{\partial t}\mathcal{E}_{X_f}(g_t) = i\langle 2i(i\nabla_f)g_t, g_t\rangle = -2\mathcal{E}_{i\nabla_f}(g_t)$$

Next, we show that momentum is conserved:

$$\frac{\partial}{\partial t}\mathcal{E}_{i\nabla_f}(g_t) = i\langle[\Delta, i\nabla_f]g_t, g_t\rangle$$

Since $[\Delta, i\nabla_f] = [-\frac{\partial^2}{\partial x^2}, i\frac{\partial}{\partial x}] = 0$

$$\frac{\partial}{\partial t}\mathcal{E}_{i\nabla_f}(g_t) = 0$$

Thus $\mathcal{E}_{i\nabla_f}(g_t) = \mathcal{E}_{i\nabla_f}(g_0)$ for all $t$. Integrating the position equation

$$\mathcal{E}_{X_f}(g_t) = \mathcal{E}_{X_f}(g_0) + \int_0^t(-2\mathcal{E}_{i\nabla_f}(g_0))ds = \mathcal{E}_{X_f}(g_0) - 2t\mathcal{E}_{i\nabla_f}(g_0)$$

$\square$

For real-valued signals, the expected location remains constant under Schrödinger evolution, which motivates the need for modulation to achieve directional transport.

**Theorem B.3** (Linear Evolution of Expected Feature in the Classical Case). *Given two real valued signals $g, h$ such that $g$ is modulated by $h$ at the initial state $g_0 = D_{i\theta h}g$, the evolution of the expected feature is*

$$\mathcal{E}_{X_f}(g_t) = \mathcal{E}_{X_f}(g) - t\theta\int h'(x)|g(x)|^2 dx$$

*Proof.* Using the basic evolution from Theorem B.2 and that expected location is invariant to modulation:

$$\mathcal{E}_{X_f}(g_t) = \mathcal{E}_{X_f}(g_0) + t\mathcal{E}_{i\nabla_f}(g_0) = \mathcal{E}_{X_f}(g) + t\mathcal{E}_{i\nabla_f}(g_0)$$

Isolating the expected momentum:

$$t\mathcal{E}_{i\nabla_f}(g_0) = ti \int \overline{g(x)e^{i\theta h(x)}} \frac{d}{dx}(g(x)e^{i\theta h(x)})dx = ti \int \overline{g(x)}e^{-i\theta h(x)}(g'(x)e^{i\theta h(x)}+i\theta h'(x)g(x)e^{i\theta h(x)})dx$$

$$= t\mathcal{E}_{i\nabla_f}(g) - t\theta \int h'(x)|g(x)|^2 dx$$

Substituting back into the equation:

$$\mathcal{E}_{X_f}(g_t) = \mathcal{E}_{X_f}(g) - t\theta \int h'(x)|g(x)|^2 dx$$

$\square$

**Theorem B.4** (Real Signals Have Constant Expected Position). *For any real-valued signal $g : \mathbb{R} \to \mathbb{R}$, the expected position remains constant under Schrödinger evolution:*

$$\mathcal{E}_{X_f}(g_t) = \mathcal{E}_{X_f}(g) \quad \text{for all } t$$

*Proof.* From Theorem B.2, $\mathcal{E}_{X_f}(g_t) = \mathcal{E}_{X_f}(g) - 2t\mathcal{E}_{i\nabla_f}(g)$. For real-valued $g$, we have $\mathcal{E}_{i\nabla_f}(g) = 0$ since $\langle g, i\nabla_f g \rangle = -i \int g(x)g'(x)dx = 0$. Therefore $\mathcal{E}_{X_f}(g_t) = \mathcal{E}_{X_f}(g)$. $\square$

**Theorem B.5** (Time Derivative of Position Variance in the Free Schrödinger Case). *Let $g \in L^2(\mathbb{R})$ be a normalized wavefunction, and let $g_t = e^{-it\Delta} g$ denote the free Schrödinger evolution with $\Delta = -\nabla_f^2$. Then the time derivative of the variance of position is:*

$$\frac{\partial}{\partial t}\mathcal{V}_{X_f}(g_t) = \mathcal{E}_{i[\Delta, X_f^2]}(g_t) + 4\left(\mathcal{E}_{X_f}(g) - 2t\,\mathcal{E}_{i\nabla_f}(g)\right)\mathcal{E}_{i\nabla_f}(g)$$

*Proof of Theorem B.5.* The variance of $X_f$ at time $t$ is:

$$\mathcal{V}_{X_f}(g_t) = \mathcal{E}_{X_f^2}(g_t) - \mathcal{E}_{X_f}(g_t)^2.$$

Differentiating with respect to $t$ and using the free particle result $\mathcal{E}_{X_f}(g_t) = \mathcal{E}_{X_f}(g) - 2t\,\mathcal{E}_{i\nabla_f}(g)$ and that the time derivative of the expected position equals the expected momentum (with our conventions $\frac{\partial}{\partial t}\mathcal{E}_{X_f}(g_t) = -2\,\mathcal{E}_{i\nabla_f}(g)$):

$$\frac{\partial}{\partial t}\mathcal{V}_{X_f}(g_t) = \frac{\partial}{\partial t}\mathcal{E}_{X_f^2}(g_t) - 2\,\mathcal{E}_{X_f}(g_t) \cdot \frac{\partial}{\partial t}\mathcal{E}_{X_f}(g_t) = \frac{\partial}{\partial t}\mathcal{E}_{X_f^2}(g_t) + 4\,\mathcal{E}_{X_f}(g_t)\,\mathcal{E}_{i\nabla_f}(g).$$

Under unitary Schrödinger evolution, for any observable $A$:

$$\frac{\partial}{\partial t}\mathcal{E}_A(g_t) = \mathcal{E}_{i[\Delta, A]}(g_t).$$

Thus, substituting $A = X_f^2$ and $\mathcal{E}_{X_f}(g_t) = \mathcal{E}_{X_f}(g) - 2t\,\mathcal{E}_{i\nabla_f}(g)$ yields

$$\frac{\partial}{\partial t}\mathcal{V}_{X_f}(g_t) = \mathcal{E}_{i[\Delta, X_f^2]}(g_t) + 4\left(\mathcal{E}_{X_f}(g) - 2t\,\mathcal{E}_{i\nabla_f}(g)\right)\mathcal{E}_{i\nabla_f}(g).$$

$\square$

## C  SCHRÖDINGER DYNAMICS

**Theorem C.1** (Expected Momentum Conservation). *For the Schrödinger evolution $g_t = \mathcal{S}_t g$, the expected momentum is conserved:*

$$\mathcal{E}_{i\nabla_f}(g_t) = \mathcal{E}_{i\nabla_f}(g) \quad \text{for all } t$$

*Proof of Theorem 3.4.* We showed previously that the Schrödinger operator is unitary and that it commutes with $\nabla_f$ because it is represented by a sum of identity matrices and powers of $\nabla_f$ itself, thus we can say:

$$\mathcal{E}_{i\nabla_f}(\mathcal{S}_t g) = \langle i\nabla_f \mathcal{S}_t g, \mathcal{S}_t g \rangle = \langle i\mathcal{S}_{-t}\nabla_f \mathcal{S}_t g, g \rangle = \langle i\nabla_f g, g \rangle$$

$\square$

**Definition C.2** ($\epsilon - f$ Regular Signal). *Let $G = (V, E)$ be a graph, $f : V \to \mathbb{R}$ be a signal, and $W_f$ be the $f$-smoothing operator, a signal $g : V \to \mathbb{C}$ is called $\epsilon - f$ regular if there exists a signal $e_g$ such that*

$$W_f g = g + e_g, \quad \|e_g\|_2 \leq \epsilon$$

**Lemma C.3** (Smoothing Operator as Commutator).

$$W_f = -i[\nabla_f, X_f] \ = \ -i(\nabla_f X_f - X_f \nabla_f)$$

*Proof.* For any signal $g$ and vertex $v$:

$$
\begin{aligned}
([\nabla_f, X_f]g)(v) &= (\nabla_f X_f g)(v) - (X_f \nabla_f g)(v) \\
&= i \sum_{w \in V} a_{v,w}(f(w) - f(v))f(w)g(w) - f(v) \cdot i \sum_{w \in V} a_{v,w}(f(w) - f(v))g(w) \\
&= i \sum_{w \in V} a_{v,w}(f(w) - f(v))^2 g(w) = i(W_f g)(v)
\end{aligned}
$$

Therefore $W_f = -i[\nabla_f, X_f]$. $\qquad\square$

**Lemma C.4** (Commutator Expansion for Schrödinger Laplacian). *For the Schrödinger Laplacian $\Delta = -\nabla_f^2$ and feature operator $X_f$, we have:*

$$i[\Delta, X_f] = -i\nabla_f W_f - iW_f \nabla_f$$

*where $W_f = [\nabla_f, X_f]$ is the $f$-smoothing operator.*

*Proof.* Using the product rule for commutators $[AB, C] = A[B, C] + [A, C]B$, we have:

$$
\begin{aligned}
i[\Delta, X_f] = i[-\nabla_f^2, X_f] &= -i[\nabla_f^2, X_f] = -i[\nabla_f \nabla_f, X_f] \\
&= -i\nabla_f[\nabla_f, X_f] - i[\nabla_f, X_f]\nabla_f \\
&= -i\nabla_f W_f - iW_f \nabla_f
\end{aligned}
$$

$\qquad\square$

*Proof of Theorem 3.6.* We start from the limit definition of the time derivative:

$$\frac{\partial}{\partial t}\mathcal{E}_{X_f}(g_t) = \lim_{h \to 0} \frac{\mathcal{E}_{X_f}(g_{t+h}) - \mathcal{E}_{X_f}(g_t)}{h}.$$

Because $g_{t+h} = \mathcal{S}_h g_t$ and $\mathcal{S}_t$ is unitary, we may write

$$\mathcal{E}_{X_f}(g_{t+h}) = \langle X_f \mathcal{S}_h g_t, \mathcal{S}_h g_t \rangle = \langle \mathcal{S}_{-h} X_f \mathcal{S}_h g_t, g_t \rangle.$$

Using the Hadamard lemma $\mathcal{S}_{-h} X_f \mathcal{S}_h = X_f + h\, i[\Delta, X_f] + o(h)$ we obtain

$$
\begin{aligned}
\mathcal{E}_{X_f}(g_{t+h}) - \mathcal{E}_{X_f}(g_t) &= \langle h\, i[\Delta, X_f]g_t, g_t \rangle + o(h) \\
&= h\, \langle i[\Delta, X_f]g_t, g_t \rangle + o(h).
\end{aligned}
$$

Dividing by $h$ and taking $h \to 0$ gives

$$\frac{\partial}{\partial t}\mathcal{E}_{X_f}(g_t) = \langle i[\Delta, X_f]g_t, g_t \rangle.$$

Substituting $\Delta = -\nabla_f^2$ and using Lemma C.4 yields

$$\frac{\partial}{\partial t}\mathcal{E}_{X_f}(g_t) = -\big(\langle i\nabla_f W_f g_t, g_t \rangle + \langle W_f i\nabla_f g_t, g_t \rangle\big)$$

$i\nabla_f$ is hermitian

$$= -\big(\langle W_f g_t, i\nabla_f g_t \rangle + \langle i\nabla_f g_t, W_f g_t \rangle\big)$$

$$= -\big(\overline{\langle i\nabla_f g_t, W_f g_t \rangle} + \langle i\nabla_f g_t, W_f g_t \rangle\big) = -2\operatorname{Re}\big(\langle i\nabla_f g_t, W_f g_t \rangle\big)$$

where we used the fact that $W_f$ is self-adjoint, the properties of inner products, and the identities $\operatorname{Re}(z) = \frac{z+\bar{z}}{2}$ and $\operatorname{Im}(z) = \frac{z-\bar{z}}{2i} = -i\operatorname{Re}(iz)$. $\qquad\square$

*Proof of Theorem 3.8.* For the modulated signal $D_{\theta h} g(v) = g(v)e^{i\theta h(v)}$:

$$(\nabla_f D_{\theta h} g(m) = i \sum_{n \in V} a_{m,n} g(n) e^{i\theta h(n)} (f(n) - f(m))$$

The expected momentum is:

$$\begin{aligned}
\mathcal{E}_{i\nabla_f}(D_{\theta h} g) &= \langle i\nabla_f D_{\theta h} g, D_{\theta h} g \rangle \\
&= \sum_{m \in V} \overline{g(m)e^{i\theta h(m)}} \cdot i \sum_{n \in V} a_{m,n} g(n) e^{i\theta h(n)} (f(n) - f(m)) \\
&= i \sum_{m \in V} \sum_{n \in V} a_{m,n} g(m) g(n) e^{i\theta(h(n) - h(m))} (f(n) - f(m))
\end{aligned}$$

Using the symmetry of undirected graphs and Euler's formula $e^{i\theta} = \cos(\theta) + i\sin(\theta)$:

$$\begin{aligned}
\mathcal{E}_{i\nabla_f}(D_{\theta h} g) &= i \sum_{(m,n) \in E} a_{m,n} g(m) g(n) [e^{i\theta(h(n) - h(m))} (f(n) - f(m)) + e^{i\theta(h(m) - h(n))} (f(m) - f(n))] \\
&= i \sum_{(m,n) \in E} a_{m,n} g(m) g(n) (f(n) - f(m)) [e^{i\theta(h(n) - h(m))} - e^{-i\theta(h(n) - h(m))}] \\
&= i \sum_{(m,n) \in E} a_{m,n} g(m) g(n) (f(n) - f(m)) \cdot 2i \sin(\theta(h(n) - h(m))) \\
&= -2 \sum_{(m,n) \in E} a_{m,n} g(m) g(n) (f(n) - f(m)) \sin(\theta(h(n) - h(m)))
\end{aligned}$$

$\square$

**Theorem C.5** (Deviation Bounds for expected feature Dynamics). *For the Schrödinger operator $\mathcal{S}_t = e^{-it\Delta}$ with $\Delta = -\nabla_f^2$ and signal $g : V \to \mathbb{C}$, if its evolved form $g_t = \mathcal{S}_t g$ is $\epsilon$-$f$ regular, the deviation between the time derivative of expected feature and the expected momentum is bounded:*

$$\left| \frac{\partial}{\partial t} \mathcal{E}_{X_f}(g_t) - \mathcal{E}_{i\nabla_f}(g) \right| \leq 2\epsilon \|\nabla_f\|_{op} \|g\|_2$$

*Proof of Theorem C.5.* recall from 3.6 that

$$\frac{\partial}{\partial t} \mathcal{E}_{X_f}(g_t) = -2\mathrm{Re}\Big( \langle i\nabla_f g_t, W_f g_t \rangle \Big).$$

By the $\epsilon$-$f$ regularity assumption there exists $e_{g_t}$ with $\|e_{g_t}\|_2 \leq \epsilon$ such that $W_f g_t = g_t + e_{g_t}$. Substituting this identity gives

$$\left| \frac{\partial}{\partial t} \mathcal{E}_{X_f}(g_t) + 2\,\mathcal{E}_{i\nabla_f}(g_t) \right| = \left| -2\,\mathrm{Re}\Big( \langle i\nabla_f g_t, e_{g_t} \rangle \Big) \right|$$

$$\leq 2\,\|i\nabla_f g_t\|_2 \|e_{g_t}\|_2 \leq 2\epsilon \,\|\nabla_f\|_F \|g_t\|_2 = 2\epsilon\,\|\nabla_f\|_F \|g\|_2$$

$\square$

*Proof of Expected multi-Feature Derivative Theorem 3.9.* To prove the theorem, we start by considering the limit definition of the time derivative of the expected feature:

$$\frac{\partial}{\partial t} \mathcal{E}_{X_{f_k}}(g_t) = \lim_{h \to 0} \frac{\langle X_{f_k} g_{t+h}, g_{t+h} \rangle - \langle X_{f_k} g_t, g_t \rangle}{h}$$

Since $g_{t+h} = \mathcal{S}_h g_t$ and $\mathcal{S}_h = e^{-ih\Delta}$ is unitary, we have:

$$\begin{aligned}
\langle X_{f_k} g_{t+h}, g_{t+h} \rangle &= \langle X_{f_k} \mathcal{S}_h g_t, \mathcal{S}_h g_t \rangle \\
&= \langle \mathcal{S}_h^* X_{f_k} \mathcal{S}_h g_t, g_t \rangle \\
&= \langle \mathcal{S}_{-h} X_{f_k} \mathcal{S}_h g_t, g_t \rangle
\end{aligned}$$

Using the expansion $\mathcal{S}_h = I - ih\Delta + o(h^2)$ and $\mathcal{S}_{-h} = I + ih\Delta + o(h^2)$, we compute:

$$\mathcal{S}_{-h} X_{f_k} \mathcal{S}_h = (I + ih\Delta + o(h^2)) X_{f_k} (I - ih\Delta + o(h^2))$$

$$= X_{f_k} + ih\Delta X_{f_k} - ih X_{f_k} \Delta + o(h^2)$$

$$= X_{f_k} + ih[\Delta, X_{f_k}] + o(h^2)$$

Therefore:

$$\frac{\langle \mathcal{S}_{-h} X_{f_k} \mathcal{S}_h g_t, g_t \rangle - \langle X_{f_k} g_t, g_t \rangle}{h} = \frac{\langle (X_{f_k} + ih[\Delta, X_{f_k}] + o(h^2)) g_t, g_t \rangle - \langle X_{f_k} g_t, g_t \rangle}{h}$$

$$= i\langle [\Delta, X_{f_k}] g_t, g_t \rangle + o(h)$$

Taking the limit as $h \to 0$:

$$\frac{\partial}{\partial t} \mathcal{E}_{X_{f_k}}(g_t) = \lim_{h \to 0} i\langle [\Delta, X_{f_k}] g_t, g_t \rangle + o(h) = \langle i[\Delta, X_{f_k}] g_t, g_t \rangle$$

$$= - \sum_j \left\langle [i\nabla_{f_j}^2, X_{f_k}] g_t, g_t \right\rangle$$

$$= -2\mathrm{Im} \left\langle i\nabla_{f_k} g_t, W_{f_k} g_t \right\rangle + \sum_{j \neq k} \left\langle [i\nabla_{f_j}^2, X_{f_k}] g_t, g_t \right\rangle.$$

This completes the proof. $\qquad \square$

**Theorem C.6** (Multi Channel Deviation Bounds for expected feature Dynamics). *For the Schrödinger operator $\mathcal{S}_t = e^{-it\Delta}$, the deviation between the time derivative of expected feature and the expected momentum is bounded as follows: For signals $\{f_1, \ldots, f_N\}$ forming a $\delta$-Position-Momentum Commuting set, and $g_t = \mathcal{S}_t g$ being $\epsilon$-$f_k$ regular for each $k$, with $\Delta = - \sum_{n=1}^N \nabla_{f_n}^2$:*

$$\left| \frac{\partial}{\partial t} \mathcal{E}_{X_{f_k}}(g_t) - 2\mathcal{E}_{i\nabla_{f_k}}(g) \right| \leq 2\epsilon \|\nabla_{f_k}\|_{op} \|g\|_2 + \delta \sum_{j \neq k} 2\|\nabla_{f_j}\|_{op} \|g\|_2^2$$

*Proof of Theorem C.6.* Using Theorem 3.9,

$$\frac{\partial}{\partial t} \mathcal{E}_{X_{f_k}}(g_t) = \langle i[\Delta, X_{f_k}] g_t, g_t \rangle = - \sum_{n=1}^N \langle i[\nabla_{f_n}^2, X_{f_k}] g_t, g_t \rangle$$

We split the sum into the $n = k$ term and the cross terms $n \neq k$:

$$\frac{\partial}{\partial t} \mathcal{E}_{X_{f_k}}(g_t) = -\langle i[\nabla_{f_k}^2, X_{f_k}] g_t, g_t \rangle - \sum_{n \neq k} \langle i[\nabla_{f_n}^2, X_{f_k}] g_t, g_t \rangle$$

For the main term ($n = k$), by the single-feature deviation bound (Theorem C.5):

$$\left| -\langle i[\nabla_{f_k}^2, X_{f_k}] g_t, g_t \rangle - 2\mathcal{E}_{i\nabla_{f_k}}(g) \right| \leq 2\epsilon \|\nabla_{f_k}\|_{op} \|g\|_2$$

Note that in the multi-feature case, $\mathcal{E}_{i\nabla_{f_k}}(g_t)$ may not be exactly constant, but we compare to the initial value $\mathcal{E}_{i\nabla_{f_k}}(g)$.

For each cross term $n \neq k$, using the $\delta$-commuting property $[X_{f_k}, \nabla_{f_n}] = E_{k,n}$ with $\|E_{k,n}\|_{op} \leq \delta$, we expand:

$$[\nabla_{f_n}^2, X_{f_k}] = \nabla_{f_n}[\nabla_{f_n}, X_{f_k}] + [\nabla_{f_n}, X_{f_k}]\nabla_{f_n} = -(\nabla_{f_n} E_{k,n} + E_{k,n}\nabla_{f_n})$$

Thus,

$$|\langle i[\nabla_{f_n}^2, X_{f_k}] g_t, g_t \rangle| = |\langle i(-\nabla_{f_n} E_{k,n} - E_{k,n}\nabla_{f_n}) g_t, g_t \rangle| \leq 2\delta \|\nabla_{f_n}\|_{op} \|g_t\|_2^2 = 2\delta \|\nabla_{f_n}\|_{op} \|g\|_2^2$$

Summing over $n \neq k$:

$$\left| \sum_{n \neq k} \langle i[\nabla_{f_n}^2, X_{f_k}] g_t, g_t \rangle \right| \leq \delta \sum_{n \neq k} 2\|\nabla_{f_n}\|_{op} \|g\|_2^2$$

Combining both parts:

$$\left|\frac{\partial}{\partial t}\mathcal{E}_{X_{f_k}}(g_t) - 2\mathcal{E}_{i\nabla_{f_k}}(g)\right| \leq 2\epsilon\|\nabla_{f_k}\|_{op}\|g\|_2 + \delta\sum_{j\neq k}2\|\nabla_{f_j}\|_{op}\|g\|_2^2$$

$\square$

*Proof of the Variance Dynamics Theorem 3.12.* Starting from the definition of variance:

$$\mathcal{V}_{X_f}(g_t) = \mathcal{E}_{X_f^2}(g_t) - \mathcal{E}_{X_f}(g_t)^2$$

Taking the derivative with respect to $t$:

$$\frac{\partial}{\partial t}\mathcal{V}_{X_f}(g_t) = \frac{\partial}{\partial t}\mathcal{E}_{X_f^2}(g_t) - 2\mathcal{E}_{X_f}(g_t)\frac{\partial}{\partial t}\mathcal{E}_{X_f}(g_t)$$

From the time evolution of expected feature for every observable, we know that:

$$\frac{\partial}{\partial t}\mathcal{E}_{X_f^2}(g_t) = \mathcal{E}_{i[\Delta, X_f^2]}(g_t)$$

Substituting this into our expression:

$$\frac{\partial}{\partial t}\mathcal{V}_{X_f}(g_t) = \mathcal{E}_{i[\Delta, X_f^2]}(g_t) - 2\mathcal{E}_{X_f}(g_t)\mathcal{E}_{i[\Delta, X_f]}(g_t)$$

using theorem 3.6

$$= \mathcal{E}_{i[\Delta, X_f^2]}(g_t) + 4\mathcal{E}_{X_f}(g_t)\mathrm{Re}\Big(\langle i\nabla_f g_t, W_f g_t\rangle\Big)$$

$\square$

*Proof of the Mixed Derivative of The Signal Routing Measure Claim 3.13 .*

$$\frac{d}{dt}\mathcal{P}_{X_f}(g, g_t, r)|_{t=0} = \frac{d}{dt}\frac{\mathcal{V}_{X_f}(g_t) + (r - \mathcal{E}_{X_f}(g_t))^2}{\mathcal{V}_{X_f}(g)}|_{t=0}$$

$$= \frac{\mathcal{E}_{i[\Delta, X_f^2]}(g_0) + 4\mathcal{E}_{X_f}(g_0)\mathrm{Re}\big(\langle i\nabla_f g_0, W_f g_0\rangle\big) - 2(r - \mathcal{E}_{X_f}(g_0))\frac{d}{dt}\mathcal{E}_{X_f}(g_t)|_{t=0}}{\mathcal{V}_{X_f}(g)|_{t=0}}$$

using the 3.6

$$= \frac{\mathcal{E}_{i[\Delta, X_f^2]}(g_0) + 4\mathcal{E}_{X_f}(g_0)\mathrm{Re}\big(\langle i\nabla_f g_0, W_f g_0\rangle\big) + 4(r - \mathcal{E}_{X_f}(g_0))\mathrm{Re}\big(\langle i\nabla_f g_0, W_f g_0\rangle\big)}{\mathcal{V}_{X_f}(g_0)}$$

$$= \frac{\mathcal{E}_{i[\Delta, X_f^2]}(g_0) + 4r\mathrm{Re}\big(\langle i\nabla_f g_0, W_f g_0\rangle\big)}{\mathcal{V}_{X_f}(g_0)}$$

Treating the measure derivative at $t = 0$ as a function of $\theta$ we get

$$\frac{\mathcal{E}_{i[\Delta, X_f^2]}(D_{\theta h}g) + 4r\mathrm{Re}\big(\langle i\nabla_f D_{\theta h}g, W_f D_{\theta h}g\rangle\big)}{\mathcal{V}_{X_f}(D_{\theta h}g)}$$

Taking the derivative with respect to $\theta$ to show that for nontrivial signals when $\theta = 0$ the value of the derivative is nonzero, thus the use of modulation can minimize the measure value

$$\frac{d}{d\theta}\frac{\mathcal{E}_{i[\Delta, X_f^2]}(D_{\theta h}g) + 4r\mathrm{Re}\big(\langle i\nabla_f D_{\theta h}g, W_f D_{\theta h}g\rangle\big)}{\mathcal{V}_{X_f}(D_{\theta h}g)} =$$

$$= \frac{\frac{d}{d\theta}\mathcal{E}_{i[\Delta, X_f^2]}(D_{\theta h}g) + 4r\frac{d}{d\theta}\mathrm{Re}\big(\langle i\nabla_f D_{\theta h}g, W_f D_{\theta h}g\rangle\big)}{\mathcal{V}_{X_f}(g)}$$

We can interpret $D_{\theta h}g = e^{i\theta X_h}g = (I + i\theta X_h + o(\theta^2))g$

$$\frac{d}{d\theta}\mathcal{E}_{i[\Delta, X_f^2]}(D_{\theta h}g) = \lim_{\epsilon \to 0}\frac{\mathcal{E}_{i[\Delta, X_f^2]}(D_{(\theta+\epsilon)h}g) - \mathcal{E}_{i[\Delta, X_f^2]}(D_{\theta h}g)}{\epsilon}$$

$$= \lim_{\epsilon \to 0}\frac{\langle i(I - i\epsilon X_h + o(\theta^2)[\Delta, X_f^2](I + i\epsilon X_h + o(\theta^2)D_{\theta h}g, D_{\theta h}g\rangle - \mathcal{E}_{i[\Delta, X_f^2]}(D_{\theta h}g)}{\epsilon}$$

$$= -\langle[[\Delta, X_f^2], X_h]D_{\theta h}g, D_{\theta h}g\rangle$$

At $\theta = 0$:

$$\frac{d}{d\theta}\mathcal{E}_{i[\Delta, X_f^2]}(D_{\theta h}g)|_{\theta=0} = -\langle[[\Delta, X_f^2], X_h]g, g\rangle$$

For the second term, we use the fact that for $F(\theta) = \langle i\nabla_f e^{i\theta X_h}g, W_f e^{i\theta X_h}g\rangle$:

$$\frac{d}{d\theta}\mathrm{Re}(F(\theta)) = \mathrm{Re}\left(\frac{d}{d\theta}F(\theta)\right)$$

Computing the derivative:

$$\frac{d}{d\theta}\langle i\nabla_f e^{i\theta X_h}g, W_f e^{i\theta X_h}g\rangle = \langle i\nabla_f(iX_h)e^{i\theta X_h}g, W_f e^{i\theta X_h}g\rangle + \langle i\nabla_f e^{i\theta X_h}g, W_f(iX_h)e^{i\theta X_h}g\rangle$$

$$= -\langle\nabla_f X_h e^{i\theta X_h}g, W_f e^{i\theta X_h}g\rangle - i\langle i\nabla_f e^{i\theta X_h}g, W_f X_h e^{i\theta X_h}g\rangle$$

At $\theta = 0$:

$$\left.\frac{d}{d\theta}F(\theta)\right|_{\theta=0} = -\langle\nabla_f X_h g, W_f g\rangle + \langle\nabla_f g, W_f X_h g\rangle$$

$$\left.\frac{\partial}{\partial\theta}\frac{\partial}{\partial t}\mathcal{P}_{X_f}(g, \mathcal{S}[t, f]D[\theta h]g, r)\right|_{t=\theta=0} = \frac{\langle[X_h, [\Delta, X_f^2]]g, g\rangle\ +\ 4r\,\mathrm{Re}\,\langle[X_h, W_f\nabla_f]g, g\rangle}{\mathcal{V}_{X_f}(g)}.$$

This completes the proof.

$\square$

# D  PROPERTIES OF UNITARY OPERATORS ON GRAPHS

In a general Hilbert space $\mathcal{H}_G$ of graph signals, a **unitary operator** $U : \mathcal{H}_G \to \mathcal{H}_G$ satisfies $U^*U = UU^* = I$. Unitary operators generated by self-adjoint operators, such as the Schrödinger operator $\mathcal{S}_t = e^{-it\Delta}$ where $\Delta$ is self-adjoint, possess several fundamental properties that make them particularly suitable for graph neural network applications. We establish these properties formally below.

**Theorem D.1** (Inner Product Preservation). *A unitary operator $U$ preserves the inner product structure of the Hilbert space. For any two signals $f, g : V \to \mathbb{C}$*

$$\langle Uf, Ug\rangle = \langle f, g\rangle$$

The inner product preservation ensures norm preservation: $\|Uf\| = \|f\|$ for any signal $f$, which guarantees numerical stability during the evolution process, preventing signal amplification or attenuation that could lead to vanishing or exploding gradients in deep network architectures.

**Theorem D.2** (Equivariance). *Let $P$ be a permutation matrix corresponding to a graph automorphism. A unitary operator $U$ commutes with $P$ if it is generated by a self-adjoint operator that commutes with $P$. In particular, for the Schrödinger operator $\mathcal{S}_t = e^{-it\Delta}$ where $\Delta$ commutes with $P$, we have for any signal $f : V \to \mathbb{C}$:*

$$\mathcal{S}_t(Pf) = P(\mathcal{S}_t f)$$

*Proof of Theorem D.2.* Since $P$ is a graph automorphism, then the Laplacian commutes with $P$ (i.e., $P\Delta = \Delta P$), we have:

$$\mathcal{S}_t(Pf) = e^{-it\Delta}(Pf) = \sum_{n=0}^{\infty} \frac{(-it\Delta)^n}{n!} Pf = \sum_{n=0}^{\infty} \frac{(-it)^n}{n!} \Delta^n Pf =$$

$$= P \sum_{n=0}^{\infty} \frac{(-it)^n}{n!} \Delta^n f = P(\mathcal{S}_t f)$$

$\square$

**Theorem D.3** (Observable Conservation). *Let $A$ be a self-adjoint operator on $\mathcal{H}_G$ and $U_t = e^{itA}$ be the unitary operator generated by $A$. For any signal $f$ and any polynomial $p$, the expected value of $A$ is invariant under evolution by any unitary operator of the form $e^{itp(A)}$:*

$$\mathcal{E}_A\left(e^{itp(A)}f\right) = \mathcal{E}_A(f)$$

*In particular, for the Schrödinger operator $\mathcal{S}_t = e^{-it\Delta}$, the Dirichlet energy $\mathcal{E}_\Delta(f)$ is conserved.*

These properties establish unitary operators, and in particular the Schrödinger operator, as natural choices for information propagation on graphs while maintaining both stability and structural consistency.

*Proof of Theorem D.3.* Let $U_p = e^{itp(A)}$. We prove that $\mathcal{E}_A(U_p f) = \mathcal{E}_A(f)$:

$$\begin{aligned}
\mathcal{E}_A\left(U_p f\right) &= \langle AU_p f, U_p f \rangle \\
&= \langle U_p^* A U_p f, f \rangle \quad \text{(using unitarity of } U_p\text{)} \\
&= \langle A U_p^* U_p f, f \rangle \quad \text{(since } [A, U_p] = 0 \text{ as } U_p = e^{itp(A)}\text{)} \\
&= \langle Af, f \rangle \quad \text{(since } U_p^* U_p = I\text{)} \\
&= \mathcal{E}_A(f)
\end{aligned}$$

The key insight is that $A$ commutes with any function of $A$, including $U_p = e^{itp(A)}$. $\square$

# E IMPLEMENTATION

## E.1 MATRIX EXPONENTIAL IMPLEMENTATION

For practical implementation of the Schrödinger operator $\mathcal{S}_t = e^{-it\Delta}$, we need to compute the exponential of a matrix. We consider two common approaches:

**Taylor Series Approximation.** For an operator $A$, its exponential $e^A$ is defined through its Taylor series expansion:

$$e^A = \sum_{k=0}^{\infty} \frac{A^k}{k!} = I + A + \frac{A^2}{2!} + \frac{A^3}{3!} + \cdots$$

where $A^k$ denotes the operator $A$ applied $k$ times, and $A^0 = I$ is the identity operator. In practice, this infinite series is truncated at a finite order $T$ for computational feasibility:

$$e^A \approx \sum_{k=0}^{T} \frac{A^k}{k!}$$

For the Schrödinger operator with a small time step, this approximation provides sufficient accuracy while maintaining computational efficiency. The choice of truncation order $T$ depends on the spectral properties of the Laplacian and the desired accuracy of the evolution.

## E.2 SHIFT OPERATOR

Let $A \in \mathbb{R}^{|V| \times |V|}$ be the (symmetric) adjacency matrix with entries $a_{n,m}$ and let $f : V \to \mathbb{R}$ be a real node feature. Denote by $X_f := \mathrm{diag}(f)$ the feature-location operator. We define the graph derivative along $f$ by the Hermitian commutator

$$\nabla_f := [X_f, A] = X_f A - A X_f, \qquad (\nabla_f)_{n,m} = a_{n,m}(f(n) - f(m)).$$

This operator mixes values only across edges and measures signed change of the signal in the direction where $f$ varies. It satisfies: (i) Locality: $(\nabla_f)_{n,m} = 0$ whenever $(n, m) \notin E$. (ii) Gauge-invariance: if $f$ is constant then $\nabla_f = 0$. (iii) Structure: for real $f$ and symmetric $A$, $[X_f, A]$ is skew-symmetric, hence $\nabla_f$ is Hermitian and generates unitary dynamics. We use the feature-weighted Laplacian

$$\Delta_f := -\nabla_f^2 = -(X_f A - A X_f)^2,$$

and the unitary shift $\mathcal{S}_t = e^{-it\Delta_f}$.

## E.3 SCHRÖDINGER GNN ARCHITECTURE DETAILS

Let $f \in \mathbb{R}^{N \times K}$ denote the learned feature-location channels (after Position–Momentum Optimization), and let $X \in \mathbb{C}^{N \times J}$ be the current layer's signal. A Schr"odinger filter with $M$ terms applies

$$Y = \sum_{m=1}^{M} \mathcal{S}[t_m, f] \, D[\theta_m \, f \, T^{(m)}] \, X \, W^{(m)},$$

where $t_m, \theta_m \in \mathbb{R}$, $T^{(m)} \in \mathbb{R}^{K \times 1}$ selects a modulation direction in feature space, $W^{(m)} \in \mathbb{C}^{J \times D}$ mixes channels, and $\mathcal{S}[t, f] = e^{-it\Delta_f}$ with $\Delta_f = -\sum_k \nabla_{f_k}^2$. A typical layer stacks a nonlinearity (e.g., absolute value) and normalization after this filter, and layers are composed depth-wise. Shapes: $X \in \mathbb{C}^{N \times J}, Y \in \mathbb{C}^{N \times D}$.

**Implementation realization.** The code instantiates this design with (i) a single input modulation and (ii) a stacked unitary propagation realized via a truncated Taylor approximation. Input feature modulation (FeatureModulationLayer) given real features $X \in \mathbb{R}^{N \times d_{\mathrm{in}}}$, two linear maps $B, P \in \mathbb{R}^{d_{\mathrm{in}} \times d}$ produce

$$\tilde{X} = XB \odot \exp\left(i\, XP\right) \in \mathbb{C}^{N \times d},$$

with orthogonal initialization of $B, P$. Unitary propagation each layer approximates a unitary flow $e^{\delta \mathcal{H}}$ by a truncated series

$$\Phi_T(\mathcal{H}, \delta) \, z = \sum_{k=0}^{T} \frac{(\delta \, \mathcal{H})^k}{k!} \, z,$$

where the generator $\mathcal{H}$ is implemented by a complex GCN operator that applies an $i$-weighted aggregation. The step size $\delta$ is learned per output channel, and each layer uses a complex activation and dropout. Layers may include residual and bias.

**Position-Momentum Optimization (PMO) Implementation.** In experiments where PMO is used, we run it as a preprocessing step before training the main Schrödinger GNN. The PMO objective (Definition 3.11) is optimized via gradient descent over the training set graphs. Specifically, we initialize the linear transformation $T \in \mathbb{R}^{M \times K}$ randomly and minimize the PMO loss by iterating over batches of training graphs. For each graph, we compute the commutator norms $\|[\nabla_{f_j}^2, X_{f_i}]\|_{op}$ and the regularization term, then backpropagate to update $T$. We use the Adam optimizer with a learning rate of $10^{-3}$ and run for a fixed number of iterations (typically 50–100) until convergence. Once optimized, the transformation $T$ is fixed, and the resulting orthogonalized features $f = qT$ are used as input to the Schrödinger GNN during training and inference. This two-stage approach decouples feature orthogonalization from the main task objective, ensuring that the position and momentum operators approximately commute before learning begins.

**Complex Features.** As noted, the Schrödinger GNN operates on complex-valued features. The input features are first projected to the complex domain via the feature modulation layer described above.

**Complex Dropout and Nonlinearity.**   For dropout in complex-valued layers, we apply standard dropout only to the real part of the features while keeping the imaginary part unchanged. This preserves the phase information encoded in the imaginary component while still providing regularization. For all nonlinearities throughout the network, we apply ReLU separately to the real and imaginary parts: $\sigma(z) = \mathrm{ReLU}(\mathrm{Re}(z)) + i \cdot \mathrm{ReLU}(\mathrm{Im}(z))$, another option is using the absolute value as nonlinearity $\sigma(z) = |z|$. This component-wise approach maintains the complex structure while introducing the necessary nonlinearity for expressive power. The magnitude $|z|$ is only used at the final layer to produce real-valued outputs for downstream tasks.

**Computational Complexity.**   The primary computational cost of the Schrödinger GNN lies in the approximation of the matrix exponential $e^{-it\Delta_f}$ using the truncated Taylor series. For a truncation order $K$, this involves $K$ applications of the sparse operator $\Delta_f$ (or $\mathcal{H}$). Since $\Delta_f$ has the same sparsity pattern as the graph adjacency matrix (proportional to $|E|$ edges), each application costs $O(|E|C)$ where $C$ is the feature dimension. Thus, the total complexity per layer is $O(K|E|C)$. This is linear in the number of edges and comparable to a standard Message Passing Neural Network (MPNN) with $K$ message passing steps or a ChebNet with polynomial order $K$. In our experiments, we found $K \approx 10 - 15$ to be sufficient, making the overhead manageable compared to deep GCNs. The memory complexity is $O((|V| + |E|)C)$, similar to standard GNNs, as we do not explicitly construct the dense matrix exponential.

**Uniform Time Initialization**   We initialize the per-channel scaling parameters that modulate the Taylor steps with an independent uniform distribution. Let $C$ denote the number of output channels of a layer. We create a complex parameter $t \in \mathbb{C}^C$ and set

$$t_j \sim \mathrm{Uniform}(0, 1.5), \qquad j = 1, \ldots, C.$$

The parameter $t_j$ effectively controls the propagation distance (or time) for the $j$-th channel. By initializing these values uniformly, we enable the network to learn a diverse set of filters where some channels aggregate local information (small $t$) while others capture long-range interactions (large $t$). This design resembles a convolution operation that samples features from both close and distant nodes across different channels. When learning is disabled, a non-trainable scalar buffer with value 1.0 is used instead.

# F   EXPERIMENTS

## F.1   TOY EXPERIMENT - GRID ORTHOGONALITY

To assess the effectiveness of our optimization, we conduct a simple grid experiment. We consider a grid graph whose node features are the Cartesian coordinates $x$ and $y$. We then replace the features by $x$ and $x + y$, apply the Position–Momentum Orthogonalization optimization described earlier, and expect the learned transformation to recover two orthogonal directions. We visualize the input features and the optimized, orthogonalized features below 6.

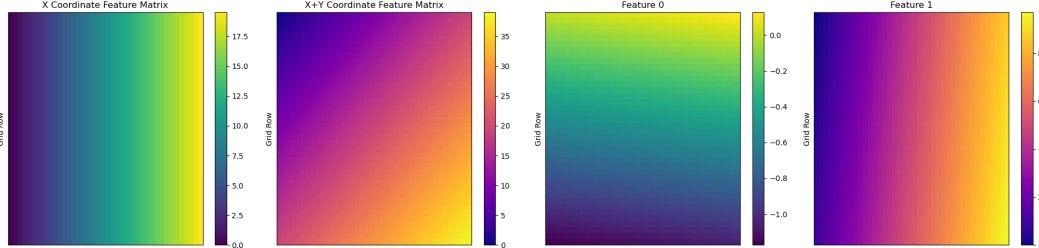

Figure 6: Grid orthogonality toy experiment. first two from left: original coordinate features $x$ and $x + y$. two to the right: features after applying the Position–Momentum Orthogonalization optimization; the recovered directions are orthogonal.

## F.2 Optimizing Signal Transport via Modulation

We constructed an experiment to show that the use of modulation can benefit signal transport on graphs. We generate $N = 60$ nodes from two 2D Gaussians, 30 around $(-1, 0)$ and 30 around $(1, 0)$ with standard deviation $0.5$ per axis. An undirected, unweighted edge is added when Euclidean distance is $< 1.5$. We define a scalar node feature $f_i$ as the $x$-coordinate, which serves as the modulation feature. We also define the $g$ graph signal as the Euclidean distance of each node's $x, y$ coordinates from $(-1, 0)$. Our target value to move the signal to is $r = 1$. We calculated the expected feature location, variance and routing measure as follows:

1. expected feature location: $\mathcal{E}_{X_f}(g) = \sum_{j=1}^{N} f_j |g_j|^2$

2. variance: $\mathcal{V}_{X_f}(g) = \mathcal{E}_{X_f^2}(g) - \mathcal{E}_{X_f}(g)^2$

3. routing measure: $\mathcal{P}_{X_f}(g_0, g_t, r) = \frac{\mathcal{V}_{X_f}(g_t) + (r - \mathcal{E}_{X_f}(g_t))^2}{\mathcal{V}\mathcal{G}_{X_f}(g_0)}$

We used our Schrödinger method $\mathcal{S}_{0.1}$, and iterated it 3 times over multiple $\theta$ values on the interval of $[-5, 5]$. Theoretically the norm should remain 1, but due to numerical instability we normalized each Schrödinger output by $\|g\|_2 = 1$ and its absolute value was taken. The results of the expected feature location, variance, and routing measure can be found in the figure 2.

## F.3 Gaussian Translate Toy Experiment

We study a controlled equivariant task on a ring graph that isolates translation behavior. Given a real signal sampled on a cycle graph, the model must learn the circular shift operator $S_d$ such that the target is $y = S_d x$. This task stresses whether a graph model can implement phase consistent transport on a simple topology.

**Data.** We generate a cycle graph with $N = 100$ nodes and undirected edges to immediate neighbors. Angles are $\theta_n = -\pi + 2\pi n/N$. For each sample we draw variance $\sigma^2 \sim \mathcal{U}[0.5, 1.5]$ (effectively bounded by "variance_random_bound=1" around the center used in code), add Gaussian noise with standard deviation $10^{-3}$, roll by a random shift, normalize to unit $\ell_2$ norm, and set the label $y = S_d x$ with $d = 35$. Datasets use an 80/10/10 split and batch size 32.

**Models.** We compare standard real-valued GNNs with Schrödinger models that implement unitary graph propagation via a truncated exponential. Let $\mathcal{A}$ denote the aggregation operator on the cycle and define the complex generator $\mathcal{H} = i\mathcal{A}$. Each Schrödinger layer applies a learnable linear map $W$ and a Taylor approximation of the unitary flow $e^{\delta\mathcal{H}}$: $z \leftarrow \sum_{k=0}^{T} \frac{(\delta\mathcal{H})^k}{k!} W z$ with $T = 15$. We use depth $L = 35$, feature normalization after every layer, and a magnitude nonlinearity. The *modulated* variants inject positional phase through a learned linear modulation direction $m = \text{Linear}([x, \theta])$ and multiply features by $e^{i\epsilon m}$ with $\epsilon = 25$. The step size $\delta$ is learnable.

**Training.** Loss is the $L_2$ distance between the model prediction $f(x_i) = \hat{y}_i$ for some sample $x_i$ and the target $y_i$, $\|\hat{y}_i - y_i\|_2$. We train with Adam (24) for 250 epochs, using two parameter groups (modulation parameters at $10\times$ the base learning rate), base learning rate $0.1$, ReduceLROnPlateau with factor $0.7$ and patience 10. The evaluation plots show smoothed test losses per epoch with a dashed reference line corresponding to a naive baseline.

**Baselines.** Vanilla GCN and GAT are trained with the same depth 35 and comparable width, using the same magnitude readout and normalization.

## F.4 MNIST Experiment Details

**Dataset Construction** The MNIST Graph dataset converts standard $28 \times 28$ pixel images into graph structures.

- **Nodes**: Each pixel is treated as a node ($N = 784$ nodes per graph).

| Model | Params |
|---|---|
| vanilla GCN | 2,136 |
| graph_attention (GAT) | 6,193 |
| Schrödinger non modulated | 4,273 |
| **Schrödinger** | 4,275 |

Table 5: Gaussian-Translate on a ring with $N = 100$ and shift $d = 35$. The modulated Schrödinger family dominates; our complex modulated model attains strong performance with substantially lower error than standard GNNs.

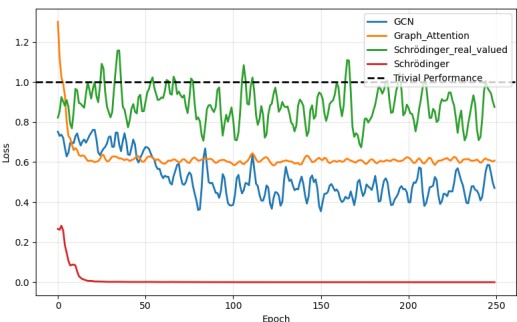

Figure 7: Gaussian-Translate learning curves. Lower is better. Our complex modulated Schrödinger model converges rapidly to the best error, outperforming real-valued and non-modulated variants, as well as standard GNN baselines. The dashed line denotes the trivial predictor.

- **Edges**: We construct an undirected graph using an 8-neighbor grid connectivity (Chebyshev radius $r = 1$), representing the local spatial structure of the image.

- **Node Features**: Each node $v_i$ is assigned a 3-dimensional feature vector $\mathbf{x}_i = [x_{norm}, y_{norm}, I]$, where $x_{norm}, y_{norm} \in [0, 1]$ are the normalized spatial coordinates and $I \in [0, 1]$ is the pixel intensity.

- **Splits**: We use the standard MNIST partition with 60,000 graphs for training and 10,000 for testing(28).

We trained each model across 5 random seeds (0-4) to report mean accuracy and standard deviation.

**Hyperparameters**

- **Hidden Dimension**: 64

- **Layers**: 3

- **Epochs**: 200

- **Batch Size**: 16

- **Optimizer**: Adam with learning rate $\alpha = 3 \times 10^{-4}$

- **Dropout**: 0.1

- **Aggregation**: Global Mean Pooling

The CNN baseline is a classical 2D convolutional neural network (27) operating directly on raw $28 \times 28$ images (not graphs). It uses the same hyperparameters (hidden dimension, number of layers, dropout, learning rate) as the GNN models, with Conv2d layers followed by adaptive average pooling and a linear classifier. This provides a non-graph reference point for comparison.

**Baselines** We evaluated five standard GNN architectures to provide a comprehensive performance benchmark:

- **GCN** (Graph Convolutional Network): Uses standard spectral graph convolution layers (25).
- **GAT** (Graph Attention Network): Utilizes attention mechanisms to learn adaptive edge weights for neighbor aggregation (53).
- **GIN** (Graph Isomorphism Network): A theoretically expressive model that uses Multi-Layer Perceptrons (MLPs) within the aggregation step to distinguish non-isomorphic graphs (56).
- **MPNN** (Message Passing Neural Network): A general framework employing explicit MLPs for both the message calculation and node update steps (16).
- **ChebConv**: A spectral graph convolution based on Chebyshev polynomials ($K = 2$), capable of approximating higher-order graph Laplacian filters to capture local geometric patterns (12).

All models use Global Mean Pooling to aggregate node embeddings into a graph-level representation for classification.

## F.5 TU EXPERIMENT - GRAPH CLASSIFICATION

This section provides a thorough explanation of the constraints and hyperparameter search process for the architecture-matched comparison presented on the datasets ENZYMES, IMDB-BINARY, MUTAG, and PROTEINS tasks from TU Dataset (36), the results can be found in table 3.

Table 6: Statistics of graph classification datasets (TU Datasets).

|  | ENZYMES | IMDB | MUTAG | PROTEINS |
|---|---|---|---|---|
| #Graphs | 600 | 1000 | 188 | 1113 |
| #Nodes (range) | 2 - 126 | 12 - 136 | 10 - 28 | 4 - 620 |
| #Edges (range) | 2 - 298 | 52 - 2498 | 20 - 66 | 10 - 2098 |
| Avg #Nodes | 32.63 | 19.77 | 17.93 | 39.06 |
| Avg #Edges | 124.27 | 193.062 | 39.58 | 145.63 |
| #Classes | 6 | 2 | 2 | 2 |
| Directed | False | False | False | False |
| ORC Mean | 0.13 | 0.58 | -0.27 | 0.17 |
| ORC Std | 0.15 | 0.19 | 0.05 | 0.20 |

**Architectural Constraints**   To ensure a fair and controlled comparison, all models were implemented with a standardized architecture consisting of six graph convolution layers followed by a single linear layer for classification. The core constraint was matching the total number of trainable parameters across all models. We first established a baseline parameter count using the Unitary (UniGCN) (23) architecture with a hidden dimension of 128. Subsequently, for all other models (GAT, GCN, GIN, Adaptive Unitary, Schrödinger, and Schrödinger PMO), we adjusted their respective hidden dimensions until their total parameter count matched the GCN baseline within a strict 0.6% tolerance. This methodology isolates the architectural differences as the primary variable, ensuring that performance variations are attributable to the intrinsic properties of the convolution operators rather than model capacity. For complex-valued models like the Schrödinger variants, each complex parameter was counted as two real-valued parameters.

**Hyperparameter Search**   We performed a grid search to identify the optimal hyperparameters for each model-dataset combination. The search space was adapted from (23) and (37) as follows:

- **Learning Rate**: {0.0005,0.001,0.005,0.01}
- **Dropout Rate**: {0,0.25,0.5}

The best-performing combination of hyperparameters was selected based on the mean validation accuracy over 100 runs fo each combination. The specific values chosen for each model are detailed in Table 7.

Table 7: Hyperparameters for the Architecture-Matched Comparison.

| MODEL | HYPERPARAMETER | ENZYMES | IMDB | MUTAG | PROTEINS |
|---|---|---|---|---|---|
| GCN | Learning Rate | 0.005 | 0.005 | 0.005 | 0.001 |
| | Dropout | 0 | 0 | 0 | 0 |
| | Hidden Dimension | 190 | 190 | 190 | 190 |
| GAT | Learning Rate | 0.001 | 0.001 | 0.0005 | 0.005 |
| | Dropout | 0 | 0.5 | 0 | 0.5 |
| | Hidden Dimension | 189 | 189 | 189 | 189 |
| Schrödinger | Learning Rate | 0.005 | 0.0005 | 0.005 | 0.005 |
| | Dropout | 0.25 | 0 | 0.25 | 0 |
| | Hidden Dimension | 117 | 170 | 170 | 170 |
| Schrödinger PMO | Learning Rate | 0.005 | 0.001 | 0.01 | 0.005 |
| | Dropout | 0 | 0 | 0 | 0 |
| | Hidden Dimension | 117 | 117 | 117 | 117 |
| Unitary | Learning Rate | 0.001 | 0.001 | 0.001 | 0.0005 |
| | Dropout | 0 | 0 | 0 | 0 |
| | Hidden Dimension | 128 | 128 | 128 | 128 |
| Adaptive Unitary | Learning Rate | 0.005 | 0.0005 | 0.005 | 0.001 |
| | Dropout | 0 | 0 | 0 | 0 |
| | Hidden Dimension | 127 | 127 | 127 | 127 |
| Adaptive Unitary PMO | Learning Rate | 0.001 | 0.01 | 0.001 | 0.001 |
| | Dropout | 0.25 | 0 | 0 | 0 |
| | Hidden Dimension | 127 | 127 | 127 | 127 |
| GIN | Learning Rate | 0.001 | 0.005 | 0.01 | 0.0005 |
| | Dropout | 0 | 0 | 0 | 0 |
| | Hidden Dimension | 190 | 190 | 190 | 190 |

**Runtime Comparison** Table 8 reports the mean and standard deviation of the training time per epoch for each model on the TU datasets.

Table 8: Runtime comparison on TU datasets (seconds per run, mean $\pm$ std).

| Model | ENZYMES | IMDB | MUTAG | PROTEINS |
|---|---|---|---|---|
| GCN | $33.4 \pm 7.85s$ | $27.5 \pm 5.66s$ | $19.0 \pm 4.30s$ | $33.9 \pm 2.02s$ |
| GAT | $61.5 \pm 15.51s$ | $42.6 \pm 0.52s$ | $14.3 \pm 2.83s$ | $65.7 \pm 8.13s$ |
| GIN | $39.8 \pm 5.41s$ | $32.9 \pm 9.12s$ | $9.1 \pm 1.52s$ | $43.2 \pm 10.40s$ |
| Unitary | $216.7 \pm 6.25s$ | $261.9 \pm 72.57s$ | $60.4 \pm 14.47s$ | $189.3 \pm 6.79s$ |
| Adaptive Unitary | $200.1 \pm 27.03s$ | $285.5 \pm 48.41s$ | $47.5 \pm 12.85s$ | $202.2 \pm 36.36s$ |
| Adaptive Unitary PMO | $84.3 \pm 15.91s$ | $142.9 \pm 7.83s$ | $45.0 \pm 0.67s$ | $158.0 \pm 0.82s$ |
| Schrödinger | $172.8 \pm 12.89s$ | $255.9 \pm 55.36s$ | $44.4 \pm 13.06s$ | $247.6 \pm 47.71s$ |
| Schrödinger PMO | $173.5 \pm 25.11s$ | $279.1 \pm 67.42s$ | $68.6 \pm 8.54s$ | $258.5 \pm 30.49s$ |

### F.5.1 DIAGNOSTIC VISUALIZATION AND MODEL VARIANTS

For empirical diagnostics, we use a variant of our Schrödinger GNN that applies phase modulation at *each layer*, where each layer derives its phase from a learned linear projection of the current layer's input features and an absolute value activation.

**Diagnostic Methodology: Expected Location and Distance.** To quantify how signal content shifts through the network, we use the expected feature location $\mathcal{E}_{X_f}(g)$ as defined in Section 3, which measures where the signal's energy is concentrated in phase space. We then define the *nor-*

*malized expected distance* for layer $l$ and channel $k$ as:

$$D_{l,k} = \frac{|\mathcal{E}_\phi(g_{\text{in}}) - \mathcal{E}_\phi(g_{\text{out}})|}{\phi_{\max} - \phi_{\min}}, \tag{8}$$

where $g_{\text{in}}$ is the broadcast (amplitude) before convolution, $g_{\text{out}}$ is the output after convolution, $\phi$ is the phase of the signal (the part in the exponent of the modulation operator), $\phi_{\max}$ and $\phi_{\min}$ are the maximum and minimum phase values in the signal. This metric captures how much the "center of mass" shifts relative to the total phase range, enabling comparison of signal transport across different layers and channels. We compare two scenarios: (i) *Conv-only*: applying only the unitary convolution without phase modulation, and (ii) *Modulation + Conv*: applying phase modulation before convolution. The diagnostic reveals that modulation systematically shifts the expected location, while conv-only operations preserve it.

**Windowed Analysis via Soft Phase Windows.** Since typical signals span the entire graph, their global expected location may not be meaningful. Following the conceptual decomposition discussed in Section 3, we partition signals into localized "chunks" using soft Gaussian windows in phase space. For channel $k$ with phase values $\phi_k(n)$, we construct $L$ windows as follows:

1. **Window centers**: Divide the phase range $[\phi_{\min}, \phi_{\max}]$ into $L$ equal regions with centers $c_l = \phi_{\min} + \frac{2l+1}{2L}(\phi_{\max} - \phi_{\min})$ for $l = 0, \ldots, L-1$.

2. **Gaussian distances**: For each node $n$ and window $l$, compute $d_l(n) = -\frac{(\phi_k(n) - c_l)^2}{2\sigma^2}$ where $\sigma = \frac{\phi_{\max} - \phi_{\min}}{2L}$.

3. **Soft partition via softmax**: Apply $w_l(n) = \frac{e^{d_l(n)}}{\sum_{l'} e^{d_{l'}(n)}}$, ensuring $\sum_l w_l(n) = 1$.

The windowed signal $g^l = w_l \odot g$ represents the portion of signal concentrated around phase center $c_l$. By tracking how each window's expected location shifts after convolution, we can visualize directional signal flow: windows in different phase regions exhibit different propagation behaviors depending on the modulation.

## F.6 HETEROPHILOUS NODE CLASSIFICATION

We evaluate our model on heterophilous node classification benchmarks from (40), which specifically test the ability of GNNs to learn on graphs where connected nodes tend to have different labels. We follow the experimental protocol from (23), using the same data splits and evaluation metrics. Results are reported in Table 9

Table 9: Performance on heterophilous node classification benchmarks. Top performing are in bold.

| TYPE | METHOD | ROMAN-E. Test AP ↑ | AMAZON-R. Test AP ↑ | MINESWEEPER ROC AUC ↑ | TOLOKERS ROC AUC ↑ | QUESTIONS ROC AUC ↑ |
|---|---|---|---|---|---|---|
| MP | GCN[†] (25) | $73.69 \pm 0.74$ | $48.70 \pm 0.63$ | $89.75 \pm 0.52$ | $83.64 \pm 0.67$ | $76.09 \pm 1.27$ |
| | SAGE[†] (19) | $85.74 \pm 0.67$ | $53.63 \pm 0.39$ | $93.51 \pm 0.57$ | $82.43 \pm 0.44$ | $76.44 \pm 0.62$ |
| | GAT[†] (53) | $80.87 \pm 0.30$ | $49.09 \pm 0.63$ | $92.01 \pm 0.68$ | $83.70 \pm 0.47$ | $77.43 \pm 1.20$ |
| | GT[†] (13) | $86.51 \pm 0.73$ | $51.17 \pm 0.66$ | $91.85 \pm 0.76$ | $83.23 \pm 0.64$ | $77.95 \pm 0.68$ |
| Unitary | Unitary GCN[‡] (23) | $87.21 \pm 0.76$ | $\mathbf{55.34 \pm 0.74}$ | $94.27 \pm 0.58$ | $84.83 \pm 0.68$ | $\mathbf{79.21 \pm 0.79}$ |
| | Lie Unitary GCN[‡] (23) | $85.50 \pm 0.22$ | $52.35 \pm 0.26$ | $96.11 \pm 0.10$ | $\mathbf{85.18 \pm 0.43}$ | $80.01 \pm 0.43$ |
| Ours | Schrödinger | $\mathbf{88.56 \pm 0.71}$ | $49.55 \pm 0.71$ | $\mathbf{96.31 \pm 0.49}$ | $84.3 \pm 0.31$ | $70.66 \pm 2.55$ |

[†]Reported performance taken from (40). [‡]Reported performance taken from (23).

**Dataset Statistics** Table 10 summarizes the statistics of the heterophilous node classification datasets.

**Experimental Setup** We follow the experimental protocol from (23). All baseline results for MP methods (GCN, SAGE, GAT, GT) are taken from (40), and Unitary GCN and Lie Unitary GCN results are taken from (23).

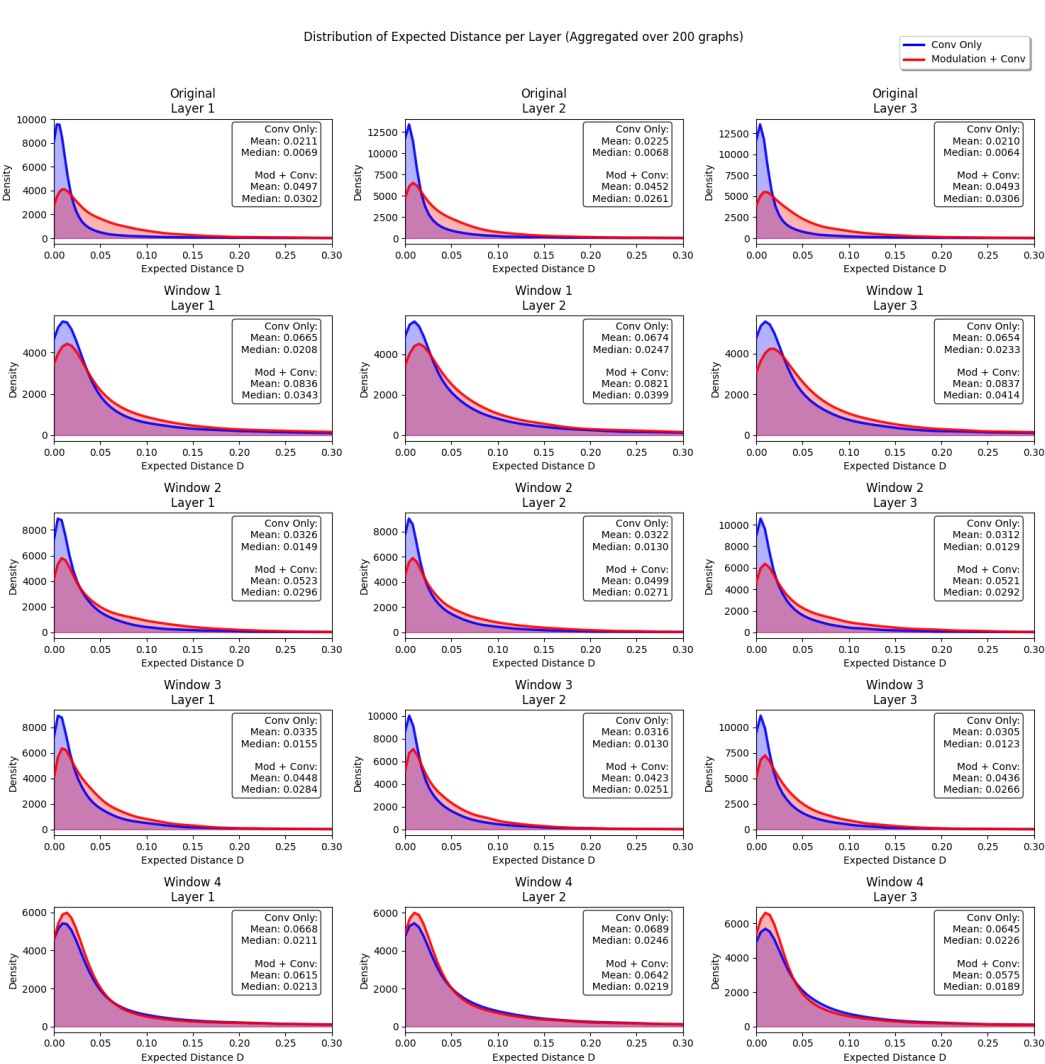

Figure 8: Distribution of expected distance $D_{l,k}$ across layers and channels. Blue curves show Conv-only, red curves show Modulation + Conv. The shift in the red distribution demonstrates that phase modulation enables directional signal transport.

Table 10: Statistics of heterophilous node classification datasets (40).

|  | ROMAN-EMPIRE | AMAZON-RATINGS | MINESWEEPER | TOLOKERS | QUESTIONS |
|---|---|---|---|---|---|
| #Nodes | 22,662 | 24,492 | 10,000 | 11,758 | 48,921 |
| #Edges | 32,927 | 93,050 | 39,402 | 519,000 | 153,540 |
| #Classes | 18 | 5 | 2 | 2 | 2 |
| Homophily | 0.05 | 0.38 | 0.68 | 0.59 | 0.84 |
| Metric | AP | AP | ROC AUC | ROC AUC | ROC AUC |

**Architecture** For our Schrödinger model, we use the following architecture:

- **Preprocessing**: Position-Momentum Optimization (PMO) run for 50 epochs with a learning rate of $0.001$ on input features to obtain orthogonalized feature locations

- **Convolution layers**: the first layer is Schrödinger layers with feature modulation and the rest are Schrödinger layer without a modulation layer

- **Readout**: Linear layer for node classification

**Hyperparameters** Table 11 shows the hyperparameter configuration for our Schrödinger model on the heterophilous benchmarks.

Table 11: Hyperparameters for Schrödinger on heterophilous node classification.

|  | ROMAN-EMPIRE | AMAZON-RATINGS | MINESWEEPER | TOLOKERS | QUESTIONS |
|---|---|---|---|---|---|
| Learning Rate | 0.001 | 0.001 | 0.001 | 0.0001 | 0.0001 |
| Dropout | 0.5 | 0.5 | 0.5 | 0.5 | 0.5 |
| # Conv. Layers | 8 | 4 | 8 | 4 | 4 |

### F.7 ABLATION STUDY ON ENZYMES

We conducted an ablation study on the ENZYMES dataset to investigate the contribution of each component in our Schrödinger GNN framework. We utilized a fixed architecture across all models: 3 graph convolution layers followed by a final linear layer, with a hidden dimension of 128, a dropout rate of 0, and a learning rate of 0.005. All models were trained for 300 epochs, and results are averaged over 100 independent trials. The ablation progression is as follows:

- **Unitary (UniGCN)**: The baseline unitary graph convolution network (23).

- **Adaptive Unitary**: Extends UniGCN by learning the time parameter $t$ in the unitary operator.

- **Schrödinger**: Further extends Adaptive Unitary by introducing feature modulation, effectively creating the full Schrödinger filter.

- **Schrödinger PMO**: The complete model which includes Position-Momentum Optimization (PMO) on the input features before applying the Schrödinger filter.

Table 12: Ablation study results on ENZYMES (Test Accuracy $\pm$ Std). All models share the same backbone architecture and hyperparameters.

| Model | Test Accuracy |
|---|---|
| Unitary (UniGCN) | $37.33 \pm 8.25$ |
| Adaptive Unitary | $41.56 \pm 5.67$ |
| Schrödinger | $43.61 \pm 4.58$ |
| Schrödinger PMO | $44.83 \pm 4.03$ |

Table 13: Statistics of Peptides datasets (LRGB). Both datasets share the same underlying graphs but differ in their prediction tasks.

| Statistic | Value |
|---|---|
| #Graphs | 15,535 |
| Avg #Nodes | 150.94 |
| Avg #Edges | 307.30 |
| TASKS | |
| PEPTIDES-FUNC | 10 (Graph Classification) |
| PEPTIDES-STRUCT | 11 (Graph Regression) |

Table 14: Hyperparameters for Schrödinger models on Peptides-Func

| | SCHRÖDINGER | SCHRÖDINGER (NON-MODULATED) |
|---|---|---|
| lr | 0.001 | 0.001 |
| dropout | 0.2 | 0.1 |
| attn dropout | 0.1 | 0.1 |
| delta init | log_stratified | log_stratified |
| # Conv. Layers | 4 | 4 |
| hidden dim. | 195 | 195 |
| node type | RSWE (42) | RSWE (42) |
| batch size | 200 | 200 |
| # epochs | 4000 | 4000 |
| edge aggregator | GINE | GINE |
| # Parameters | 493K | 492K |

## F.8 PEPTIDES

**Experimental Setup and Implementation Details**  Our evaluation framework leverages the GraphGym platform (57) for systematic assessment on Peptide datasets datasets. Tables 4 presents comprehensive benchmark results compiled from various state-of-the-art architectures, including (26; 10; 49; 47; 50; 22; 35; 54; 55; 12; 21; 6), with all reported metrics collected from published literature as of September 2025. The experimental infrastructure utilizes PyTorch (39) as the primary deep learning framework, supplemented by PyTorch Geometric (15) for specialized graph neural network operations.

**Edge Feature Handling**  A notable limitation of our unitary graph convolution implementation is the absence of native edge feature support. To address this constraint in edge-attributed datasets, we employ a preprocessing strategy incorporating either GINE (56) or Gated GCN (3) architectures as initial layers. These components serve as edge feature aggregators, effectively transforming edge attributes into node representations. When such preprocessing is utilized, we explicitly document this configuration through an "edge aggregator" hyperparameter specification in our experimental tables.

**Computational Resources and Performance**  All experimental runs were conducted on individual GPUs, specifically utilizing an NVIDIA NVIDIA L40S hardware. Training duration exhibited convergence with less than 15 seconds epochs. Dataset storage requirements was 1GB. The smaller datasets typically completed training epochs within seconds.

**Parameter Count**  LRGB datasets require a parameter limit of 500k, thus each complex parameter is count as 2.

**Hyperparameters**  We employ the Adam optimizer (24) with an initial learning rate of 0.001, utilizing a cosine learning rate scheduler and run a hyperparameter sweep for the basic model with the following hyperparameters:

Table 15: Hyperparameters for Schrödinger models on Peptides-Struct

|  | SCHRÖDINGER | SCHRÖDINGER (NON-MODULATED) |
| --- | --- | --- |
| lr | 0.001 | 0.001 |
| dropout | 0.15 | 0.1 |
| # Conv. Layers | 4 | 6 |
| hidden dim. | 150 | 64 |
| node type | LapPE (42) | LapPE (42) |
| batch size | 200 | 200 |
| # epochs | 500 | 500 |
| edge aggregator | GINE | GINE |
| # Parameters | 496K | 499K |

- **Number of layers**: $\{2, 4, 6, 8\}$
- **Dropout**: $\{0.1, 0.15, 0.2\}$
- **Hidden dimensions**: maximized according to the 500K parameter count limit and considering complex as 2 parameters.

## G    PROOFS

**Theorem G.1** (Dirichlet Energy is a Laplacian Observable). *For a signal $f$ and $\tilde{f}$ its Fourier transform, the Dirichlet energy is equivalent to the expected squared momentum in momentum space:*

$$\mathcal{E}_{\tilde{\Delta}}(f) = \frac{1}{2} \int p^2 |\tilde{f}(p)|^2 dp = \frac{1}{2} \mathcal{E}_{P^2}(\tilde{f})$$

*where $\tilde{f}(p)$ is the Fourier transform of $f$ and $p$ represents momentum.*

*Proof of Theorem G.1.* The proof follows from the spectral decomposition of the Laplacian operator:

$$\mathcal{E}_{\tilde{\Delta}}(f) = \langle \tilde{\Delta} f, f \rangle = \langle -\nabla \cdot \nabla f, f \rangle$$
$$= \frac{1}{2} \int \|\nabla f(x)\|_2^2 dx = \frac{1}{2} \int p^2 |\tilde{f}(p)|^2 dp = \frac{1}{2} \mathcal{E}_{P^2}(\tilde{f})$$

where we used Parseval's theorem and the fact that the Fourier transform of the gradient operator corresponds to multiplication by $ip$ in momentum space. $\square$

### G.1    COMMUTATOR IDENTITIES USED IN SECTION 3

We collect concise commutator expansions used in Section 3. Throughout, $X_f := \mathrm{diag}(f)$, $\nabla_f$ is as in Definition 3.1, $\Delta_f = -\nabla_f^2$, and $W_f := -i[\nabla_f, X_f]$ (Lemma: Smoothing Operator as Commutator).

**Lemma G.2** (Product-rule commutator). *For any features $f, h$,*

$$[X_h, W_f \nabla_f] = [X_h, W_f]\nabla_f + W_f[X_h, \nabla_f].$$

*Proof.* Use $[A, BC] = [A, B]C + B[A, C]$ with $A = X_h$, $B = W_f$, $C = \nabla_f$. $\square$

**Lemma G.3** (Expansion of $i[\Delta_f, X_f^2]$). *and $i[[\Delta_f, X_f^2], X_h]$] Let $S_f := X_f W_f + W_f X_f$. Then*

$$i[\Delta_f, X_f^2] = \nabla_f S_f + S_f \nabla_f.$$

*Moreover, for any feature $h$,*

$$i[[\Delta_f, X_f^2], X_h] = \nabla_f[S_f, X_h] + [\nabla_f, X_h]S_f + S_f[\nabla_f, X_h] + [S_f, X_h]\nabla_f.$$

*Since $[\nabla_f, X_h] = iW_h$ and $[S_f, X_h] = X_f[W_f, X_h] + [W_f, X_h]X_f$ (as $[X_f, X_h] = 0$ for diagonal real features), both identities reduce to products of $W_f, W_h$, and diagonal multipliers.*

*Proof.* By $[AB, C] = A[B, C] + [A, C]B$ and $[\nabla_f, X_f^2] = [\nabla_f, X_f]X_f + X_f[\nabla_f, X_f] = i(W_f X_f + X_f W_f) = iS_f$. Then

$$i[\Delta_f, X_f^2] = -i[\nabla_f^2, X_f^2] = -i\big(\nabla_f[\nabla_f, X_f^2] + [\nabla_f, X_f^2]\nabla_f\big) = \nabla_f S_f + S_f \nabla_f.$$

The double commutator follows by another application of $[AB + BA, X_h]$ and collecting terms, using $[\nabla_f, X_h] = iW_h$ and $[X_f, X_h] = 0$. $\square$

*Proof of Routing MeasureEquation* (2).

$$\mathcal{P}_A(g_0, g_t, r) = \frac{\mathcal{V}_A(g_t) + (r - \mathcal{E}_A(g_t))^2}{\mathcal{V}_A(g_0)}.$$

We will focus on the numerator of the energy flow measure, we have:

$$\langle (X_f - rI)^2 Ug, Ug \rangle = \langle (X_f^2 - 2rX_f + r^2I - \mathcal{E}_{\mathbf{X_f}}(Ug)^2 I + \mathcal{E}_{\mathbf{X_f}}(Ug)^2 I + 2\mathcal{E}_{\mathbf{X_f}}(Ug)X_f - 2\mathcal{E}_{\mathbf{X_f}}(Ug)X_f)Ug, Ug \rangle$$

Rearranging terms to complete the square:

$$= \langle (X_f^2 - 2\mathcal{E}_{\mathbf{X_f}}(Ug)X_f + \mathcal{E}_{\mathbf{X_f}}(Ug)^2 I)Ug, Ug \rangle + \langle (r^2 - \mathcal{E}_{\mathbf{X_f}}(Ug)^2)IUg, Ug \rangle + \langle 2(\mathcal{E}_{\mathbf{X_f}}(Ug) - r)X_f Ug, Ug \rangle$$

The first term is the variance:

$$\langle (X_f - \mathcal{E}_{\mathbf{X_f}}(Ug)I)^2 Ug, Ug \rangle = \mathcal{V}_{X_f}(Ug)$$

The second term simplifies using norm preservation ($\|Ug\|^2 = \|g\|^2 = 1$ for normalized signals):

$$\langle (r^2 - \mathcal{E}_{\mathbf{X_f}}(Ug)^2)IUg, Ug \rangle = (r^2 - \mathcal{E}_{\mathbf{X_f}}(Ug)^2)$$

The third term uses the definition of expected feature:

$$\langle 2(\mathcal{E}_{\mathbf{X_f}}(Ug) - r)X_f Ug, Ug \rangle = 2(\mathcal{E}_{\mathbf{X_f}}(Ug) - r)\mathcal{E}_{\mathbf{X_f}}(Ug)$$

Combining all terms:

$$\begin{aligned}
\langle (X_f - rI)^2 Ug, Ug \rangle &= \mathcal{V}_{X_f}(Ug) + (r^2 - \mathcal{E}_{\mathbf{X_f}}(Ug)^2) + 2(\mathcal{E}_{\mathbf{X_f}}(Ug) - r)\mathcal{E}_{\mathbf{X_f}}(Ug) \\
&= \mathcal{V}_{X_f}(Ug) + r^2 - \mathcal{E}_{\mathbf{X_f}}(Ug)^2 + 2\mathcal{E}_{\mathbf{X_f}}(Ug)^2 - 2r\mathcal{E}_{\mathbf{X_f}}(Ug) \\
&= \mathcal{V}_{X_f}(Ug) + r^2 + \mathcal{E}_{\mathbf{X_f}}(Ug)^2 - 2r\mathcal{E}_{\mathbf{X_f}}(Ug) \\
&= \mathcal{V}_{X_f}(Ug) + (r - \mathcal{E}_{\mathbf{X_f}}(Ug))^2
\end{aligned}$$

Therefore, the energy flow measure becomes:

$$\frac{\langle (X_f - rI)^2 Ug, Ug \rangle}{\mathcal{V}_{X_f}(g)} = \frac{\mathcal{V}_{X_f}(Ug) + (r - \mathcal{E}_{\mathbf{X_f}}(Ug))^2}{\mathcal{V}_{X_f}(g)}$$

$\square$

**Theorem G.4** (Dirichlet Energy of Feature–Modulated Signals). *Let $g : V \to \mathbb{R}$ be a real graph signal and let $h : V \to \mathbb{R}$ be a real-valued feature. For any $\theta \in \mathbb{R}$ define the modulated signal $g_\theta = D[\theta h]g$ with $D[\theta h] = \mathrm{diag}(e^{i\theta h})$. Denote the (unnormalised) graph Laplacian by $\Delta$ and its Dirichlet energy by $\mathcal{E}_\Delta(f) = \langle \Delta f, f \rangle$. Then*

$$\mathcal{E}_\Delta(g_\theta) = \mathcal{E}_\Delta(g) + \sum_{(m,n)\in E} a_{m,n}\, g(m)g(n)\big(1 - \cos\big(\theta\big(h(n) - h(m)\big)\big)\big).$$

*In particular $\mathcal{E}_\Delta(g_\theta) \geq \mathcal{E}_\Delta(g)$ with equality iff either $\theta = 0$ or $h(n) = h(m)$ for every edge $(m, n) \in E$.*

*Proof of Theorem G.4.* Recall the edge form of the Dirichlet energy $\mathcal{E}_\Delta(f) = \frac{1}{2} \sum_{(m,n)\in E} a_{m,n} |f(n) - f(m)|^2$. For $g_\theta(v) = g(v)e^{i\theta h(v)}$ we compute

$$|g_\theta(n)-g_\theta(m)|^2 = |g(n)-g(m)e^{i\theta(h(m)-h(n))}|^2 = g(n)^2+g(m)^2-2g(n)g(m)\cos\big(\theta(h(n)-h(m))\big).$$

Substituting into the edge sum gives

$$\mathcal{E}_\Delta(g_\theta) = \frac{1}{2} \sum_{(m,n)\in E} a_{m,n}\big(g(n)^2 + g(m)^2 - 2g(n)g(m)\cos(\theta\Delta h)\big)$$

$$= \mathcal{E}_\Delta(g) + \sum_{(m,n)\in E} a_{m,n}g(n)g(m)\big(1 - \cos(\theta\Delta h)\big),$$

where $\Delta h := h(n) - h(m)$. The cosine term satisfies $1 - \cos(\cdot) \geq 0$, proving the non–decreasing property and the condition for equality. $\square$

## H  LICENSES

We list below the licenses of code and datasets that we use in our experiments.

Table 16: Licenses for Code and Datasets

| MODEL/DATASET | LICENSE | NOTES |
|---|---|---|
| LRGB (14) | Custom | License |
| MNIST (28) | CC BY-SA 3.0 | Open Source |
| TUDataset (36) | Open | Open Source |
| Heterophilous Benchmarks (40) | MIT | License |
| PyTorch Geometric (15) | MIT | License |
| GraphGym (57) | MIT | License |
| GraphGPS (42) | MIT | License |
| PyTorch (39) | 3-clause BSD | License |

