# OpenReview forum: "Understanding Signal Propagation in GNNs Via Observables"
_ICLR.cc/2026/Conference — ICLR 2026 Conference Withdrawn Submission_

### Official Review · Reviewer_FfDz · 2025-10-16

**Soundness:** 2
**Presentation:** 2
**Contribution:** 2
**Rating:** 2
**Confidence:** 4

**Summary:**

This paper uses a formalism inspired by quantum mechanics to develop GNN, specifically adopting the notion of observables to examine where signals are localized, their concentration, and the effectiveness of their routing on graphs.

**Strengths:**

The introduction of quantum-mechanical style observables and related mathematical machinery seems to be novel.

**Weaknesses:**

1. The experimental evaluation is limited and overly simplistic, raising concerns about the model's ability to generalize to complex, large-scale, or highly irregular graphs. While the Schrödinger GNN demonstrates competitive performance on peptide datasets (Table 2), evaluating it on a broader range of tasks—such as node classification, heterogeneous graphs, or benchmarks from OGB/LRGB—would provide stronger empirical evidence to support its claims.

1. The proposed Schrödinger GNN lacks comprehensive ablation studies. For instance, the impact of the unitary operator choice, the necessity of feature modulation compared to alternative mechanisms, and the influence of hyperparameter settings are not thoroughly investigated.

1. Incorporating additional diagnostics, such as statistical significance tests, would enhance the reliability and confidence in the reported results.

1. Computational efficiency and practical scalability are not sufficiently addressed. Although Section 5 briefly mentions that computing matrix exponentials required by the Schrödinger operator is more expensive than classical GSO filtering, there is no quantitative analysis (e.g., wall-clock times, memory usage, or scaling with node counts) or discussion of potential optimization strategies and trade-offs. This omission leaves the practical usability of the model in larger, real-world systems uncertain.

1. The connection between the proposed observables and existing notions, such as Dirichlet energy, Cheeger constant, effective resistance, or curvature, would benefit from more comparative experiments or practical case studies. For example, can the newly introduced signal routing measure offer actionable improvements or diagnostics in scenarios where Dirichlet energy or effective resistance falls short?

1. The approach’s relationship to existing quantum-inspired GNNs and the broader literature on quantum graph neural networks is not sufficiently integrated. Unclear novelty and differentiation from prior works like "Quantum Graph Neural Networks" and "A Quantum-Inspired Neural Network for Geometric Modeling".

**Questions:**

1. A detailed analysis of the computational and memory demands for computing the matrix exponential in the Schrödinger operator is essential. How does the approach scale with increasing graph size, and what are the practical bottlenecks in real-world applications?

1. It would be valuable to identify specific classes of graphs, tasks, or data domains where the proposed approach may underperform. For example, how does it handle highly heterogeneous or irregular networks, or scenarios where meaningful formal location features are absent?

1. The utility of the proposed signal routing measure should be evaluated in terms of its correlation with improved task performance across benchmarks. Additionally, does it provide insights in cases where classical metrics like Dirichlet energy or effective resistance analysis are insufficient?

1. Additional experiments or ablation studies are needed to clarify the impact of individual design choices. For instance, the necessity of complex-valued modulation compared to purely real representations, the effect of the chosen unitary GSO, and the role of feature direction orthogonalization should be thoroughly investigated.

---

> ### Author Response · Authors · 2025-12-03
>
> >**Weaknesses:**
> >1. The experimental evaluation is limited and overly simplistic, raising concerns about the model's ability to generalize to complex, large-scale, or highly irregular graphs. While the Schrödinger GNN demonstrates competitive performance on peptide datasets (Table 2), evaluating it on a broader range of tasks—such as node classification, heterogeneous graphs, or benchmarks from OGB/LRGB—would provide stronger empirical evidence to support its claims.
>
> **Response.** Thank you for pointing this out. In the revised paper we added many more experiments and improved the hyperparameter optimization. Now, there are many experiments which show that modulation improves performance. For example TU experimet graph classification ENZYMES (40.3 ± 6.63 without modulation and 43.5 ± 4.89 with) and PROTEINS (69.19 ± 3.01 without modulation and 71.57 ± 2.56 with). **((List the experiments where modulation helps - give the performance with and without modulation))**.
> Please see Table 3 in the paper.
>
>
> >1. The proposed Schrödinger GNN lacks comprehensive ablation studies. For instance, the impact of the unitary operator choice, the necessity of feature modulation compared to alternative mechanisms, and the influence of hyperparameter settings are not thoroughly investigated.
>
> **Response.** Thank you for pointing this out. We added an ablation study to the revised paper. Please see Appendix F.7
>
> >2. Incorporating additional diagnostics, such as statistical significance tests, would enhance the reliability and confidence in the reported results.
>
> **Response.** Thank you for suggesting adding more diagnostics to our paper. We now added a new application of our theory for diagnosing the signal propagation capabilities of trained GNNs. The idea is to plot the signals of various graphs in the dataset at various layers, and see if the next GNN layer changes the location of the signal. Since signals are typically supported across the whole graph, we first multiply the signal by a window, and plug the windowed signal into the next layer, checking if the output has a translated mean location. Here, splitting allows us to see which parts of the signal propagated to which parts of the graph.
>
>
> >3. Computational efficiency and practical scalability are not sufficiently addressed. Although Section 5 briefly mentions that computing matrix exponentials required by the Schrödinger operator is more expensive than classical GSO filtering, there is no quantitative analysis (e.g., wall-clock times, memory usage, or scaling with node counts) or discussion of potential optimization strategies and trade-offs. This omission leaves the practical usability of the model in larger, real-world systems uncertain.
>
> **Response.** We note that exponential is implemented via a Taylor series, i.e., as a polynomial in the Laplacian times $i$. Hence, our method is theoretically as efficient as traditional polynomial filter methods, like ChebNet, GCN, CayleyNet, and ARMANet.
>
> We will add runtime tables to the paper, and a theoretical analysis of the efficiency of exponentiation.

---

> ### Author Response · Authors · 2025-12-03
>
> >4. The connection between the proposed observables and existing notions, such as Dirichlet energy, Cheeger constant, effective resistance, or curvature, would benefit from more comparative experiments or practical case studies. For example, can the newly introduced signal routing measure offer actionable improvements or diagnostics in scenarios where Dirichlet energy or effective resistance falls short?
>
>
> **Response.** We clearly write that our paper introduces a notion *alternative to oversmoothing and oversquashing*, for depicting the tendency of GNNs to spread the signal across the graph and lose information. This notion is based on observables and their mean and variance. Schrodinger GNNs are designed to address signal propagation in terms of observables, not in terms of oversmoothing and oversquashing.
> Please see, for example,  the abstract:
> *In this paper, we propose an alternative approach for modeling signal propagation, inspired by quantum mechanics, using the notion of observables...*
> and the introduction:
> *We aim to directly study how coherent the signal stays when it is routed between regions of the graph. .For this, we propose in the paper an alternative way to model and probe different aspects of the content of the signal and its flow. Specifically, we model...*
>
> We note that we do not claim that our notions are better than oversmoothing and oversquashing, and we do not claim that observables can analyze oversmoothing and oversquashing. The same way that oversquashing is not better than oversmoothing, and you would not try to analyze oversmoothing via the notion of oversquashing, we do not attempt to analyze oversmoothing and oversquashing via observables. We simply add another quantity to the toolkit of the deep learning practitioner.
>
> However, to partially address your question, now added a new application of our theory for diagnosing the signal propagation capabilities of trained GNNs via observables. The idea is to plot the signals of various graphs in the dataset at various layers, and see if the next GNN layer changes the location of the signal. Since signals are typically supported across the whole graph, we first multiply the signal by a window, and plug the windowed signal into the next layer, checking if the output has a translated mean location. Here, splitting allows us to see which parts of the signal propagated to which parts of the graph.
>
> >5. The approach’s relationship to existing quantum-inspired GNNs and the broader literature on quantum graph neural networks is not sufficiently integrated. Unclear novelty and differentiation from prior works like "Quantum Graph Neural Networks" and "A Quantum-Inspired Neural Network for Geometric Modeling".
>
> **Response.**  We note that the suggested papers are about quantum computing or tensor networks, while our paper is about spectral GNNs, which are in the realm of classical computing, and our method is only *inspired* by quantum mechanics.
>
> >Questions:
> >1. A detailed analysis of the computational and memory demands for computing the matrix exponential in the Schrödinger operator is essential. How does the approach scale with increasing graph size, and what are the practical bottlenecks in real-world applications?
>
> **Response.**  See our above response.
>
> >2. It would be valuable to identify specific classes of graphs, tasks, or data domains where the proposed approach may underperform. For example, how does it handle highly heterogeneous or irregular networks, or scenarios where meaningful formal location features are absent?
>
> **Response.**  We can offer the following answer: when the signal is far from being smooth in terms of the smoothing operator, Schrodinger GNNs may not be appropriate. Moreover, our new diagnostics method can help identify examples where our method indeed learns to transport the location of signals, and cases where it does not.
>
>
> >3. The utility of the proposed signal routing measure should be evaluated in terms of its correlation with improved task performance across benchmarks. Additionally, does it provide insights in cases where classical metrics like Dirichlet energy or effective resistance analysis are insufficient?
>
> **Response.**  See our above responses.
>
> >4. Additional experiments or ablation studies are needed to clarify the impact of individual design choices. For instance, the necessity of complex-valued modulation compared to purely real representations, the effect of the chosen unitary GSO, and the role of feature direction orthogonalization should be thoroughly investigated.
>
> **Response.** Thank you for pointing this out. We added an ablation study to the revised paper. Please see Appendix F.7

---

### Official Review · Reviewer_y3yf · 2025-10-26

**Soundness:** 3
**Presentation:** 2
**Contribution:** 2
**Rating:** 4
**Confidence:** 4

**Summary:**

This paper introduces a framework for analyzing how GNNs route information across a graph, drawing inspiration from quantum mechanics. The authors define three “observables” of a graph signal: (i) its location in the graph, (ii) the concentration of the signal around that location, and (iii) how much of the signal is propagated from one region to another. Using these notions, they prove that standard spectral GNNs cannot effectively move information between distant nodes (they keep the signal’s mean location fixed and only increase its spread). As a remedy, the authors propose Schrödinger GNN, a spectral GNN built with a unitary graph shift operator and complex-valued input features. A part of the feature channels encodes an abstract “formal location” and the rest encodes the actual signal. The authors show that Schrödinger filters can direct signal propagation at a roughly constant speed in any desired direction.  Empirically, they demonstrate on a synthetic ring-transport task that only the Schrödinger GNN with complex modulation is able to successfully route the signal whereas standard GCN and GAT fail.  On the LRGB peptides tasks, the Schrödinger GNN gets close to SOTA performance.

**Strengths:**

- The conceptual framework is nice. The observable-based formulation that cleanly separates oversmoothing from deliberate transport is well-motivated and the proofs clearly explain why polynomial spectral GNNs diffuse rather than route and justify the unitary/phase-modulated design.
- The toy experiment is a well-chosen ring-transport toy task that the proposed model (w/ modulation) uniquely solves. Also, the method achieves competitive results on LRGB (Peptides), which together give initial evidence of the method’s practical value.

**Weaknesses:**

- Some parts of the exposition are hard to parse, especially for a graph ML audience. It also seems to me that the paper does not explicitly discuss how oversquashing is quantitatively addressed in the observables framework, so readers may wonder how the new measures relate to known bottleneck metrics. I only found a generic oversquashing discussion in appendix A.3, which doesn’t mention Schrödinger GNNs. I would encourage the authors to clarify this connection and consider discussing oversquashing more prominently in the main body (if I have not overlooked it). Furthermore, the paper devotes substantial space to motivating the ideas from quantum mechanics, but the actual method is then only sketched very briefly (page 7, l. 360-371). I think that having a more self-contained section introducing the Schrödinger GNN would greatly improve the paper’s clarity and accessibility. For example, it is should also be made clearer throughout the paper that applying a Schrödinger GNN is a 2-step procedure including the Position-Momentum Optimization (Def. 3.11).
- While the toy example is clear, the real-graph improve the results only modestly. In Table 2, the Schrödinger GNN’s scores on Peptides-Func (AP ~71.4) and Peptides-Struct (MAE ~0.2447) are essentially matched by other GNNs. Interestingly, the impact of complex modulation on the LRGB tasks is minimal. While LRGB is the natural starting point, it has been noted to be somewhat saturated and not always diagnostic of capturing true long-range interactions [1,2]. There are several recent benchmarks that aim to better probe long-range dependencies, although it would be unreasonable to expect evaluation on these newer datasets at this point. Still, within LRGB itself, the paper’s evaluation (restricted to a single real-world dataset and only a *part* of the full suite) appears somewhat incomplete.
- Several key references are missing. For example, table 1 compares to GAT, but the corresponding citation [3] is omitted. I did not check all ~20 methods compared on the LRGB task (table 2), but, also here, key baselines that are *explicitly mentioned* (such as [4,5]) cannot be found in the references. Even if the results for these other methods were not reproduced by the authors themselves, it remains standard scientific practice to include citations to all methods mentioned in the script. I recommend the authors another careful pass over the paper.

[1] Jan Tönshoff, Martin Ritzert, Eran Rosenbluth, Martin Grohe. *Where Did the Gap Go? Reassessing the Long-Range Graph Benchmark.* TMLR 2024.

[2] Jacob Bamberger, Benjamin Gutteridge, Scott le Roux, Michael M. Bronstein, Xiaowen Dong. *On Measuring Long-Range Interactions in Graph Neural Networks.* ICML 2025.

[3] Petar Veličković, Guillem Cucurull, Arantxa Casanova, Adriana Romero, Pietro Liò, Yoshua Bengio. *Graph Attention Networks.* ICLR 2018.

[4] Benjamin Gutteridge, Xiaowen Dong, Michael Bronstein, Francesco Di Giovanni. *DRew: Dynamically Rewired Message Passing with Delay*. ICML 2023

[5] Simon Geisler, Arthur Kosmala, Daniel Herbst, Stephan Günnemann. *Spatio-Spectral Graph Neural Networks.* NeurIPS 2024.

**Questions:**

- The results suggest complex modulation has a dramatic effect on the ring task, but practically no effect on LRGB peptides (table 2). Can the authors explain under what conditions modulation would significantly help?
- What is the computational overhead of Schrödinger GNN compared to the baselines, e.g., on LRGB? How much overhead does making the signals epsilon-commuting add in general?
- As there were more unitary GNNs recently introduced, could the authors clarify conceptual differences between their method and, e.g., UniConv [6], beyond what is said in appendix A.2?

[6] Bobak T. Kiani, Lukas Fesser, and Melanie Weber. *Unitary convolutions for learning on graphs and groups.* NeurIPS 2024.

Miscellaneous: submission uses ICLR 2025 template

---

> ### Author Response · Authors · 2025-12-03
>
> >Weaknesses:
> >1. Some parts of the exposition are hard to parse, especially for a graph ML audience. It also seems to me that the paper does not explicitly discuss how oversquashing is quantitatively addressed in the observables framework, so readers may wonder how the new measures relate to known bottleneck metrics.  I only found a generic oversquashing discussion in appendix A.3, which doesn’t mention Schrödinger GNNs. I would encourage the authors to clarify this connection and consider discussing oversquashing more prominently in the main body (if I have not overlooked it).
>
> **Response.** We clearly write that our paper introduces a notion *alternative to oversmoothing and oversquashing*, for depicting the tendency of GNNs to spread the signal across the graph and lose information. This notion is based on observables and their mean and variance. Schrodinger GNNs are designed to address signal propagation in terms of observables, not in terms of oversmoothing and oversquashing.
> Please see, for example,  the abstract:
> *In this paper, we propose an alternative approach for modeling signal propagation, inspired by quantum mechanics, using the notion of observables...*
> and the introduction:
> *We aim to directly study how coherent the signal stays when it is routed between regions of the graph. .For this, we propose in the paper an alternative way to model and probe different aspects of the content of the signal and its flow. Specifically, we model...*
>
>
> >2. Furthermore, the paper devotes substantial space to motivating the ideas from quantum mechanics, but the actual method is then only sketched very briefly (page 7, l. 360-371). I think that having a more self-contained section introducing the Schrödinger GNN would greatly improve the paper’s clarity and accessibility. For example, it is should also be made clearer throughout the paper that applying a Schrödinger GNN is a 2-step procedure including the Position-Momentum Optimization (Def. 3.11).
>
> **Response.**  Thank you for this suggestion. We will write in the revised paper an appendix devoted for practitioners, which explains how Schrodinger GNNs are defined in detail.
>
>
> >3. While the toy example is clear, the real-graph improve the results only modestly. In Table 2, the Schrödinger GNN’s scores on Peptides-Func (AP ~71.4) and Peptides-Struct (MAE ~0.2447) are essentially matched by other GNNs. Interestingly, the impact of complex modulation on the LRGB tasks is minimal. While LRGB is the natural starting point, it has been noted to be somewhat saturated and not always diagnostic of capturing true long-range interactions [1,2]. There are several recent benchmarks that aim to better probe long-range dependencies, although it would be unreasonable to expect evaluation on these newer datasets at this point. Still, within LRGB itself, the paper’s evaluation (restricted to a single real-world dataset and only a part of the full suite) appears somewhat incomplete.
>
> **Response.** Thank you for pointing this out. In the revised paper we added many more experiments and improved the hyperparameter optimization. Now, there are many experiments which show that modulation improves performance. For example TU experimet graph classification ENZYMES (40.3 ± 6.63 without modulation and 43.5 ± 4.89 with) and PROTEINS (69.19 ± 3.01 without modulation and 71.57 ± 2.56 with). **((List the experiments where modulation helps - give the performance with and without modulation))**.
> Please see Table 3 in the paper.
>
> >4. Several key references are missing. For example, table 1 compares to GAT, but the corresponding citation [3] is omitted. I did not check all ~20 methods compared on the LRGB task (table 2), but, also here, key baselines that are explicitly mentioned (such as [4,5]) cannot be found in the references. Even if the results for these other methods were not reproduced by the authors themselves, it remains standard scientific practice to include citations to all methods mentioned in the script. I recommend the authors another careful pass over the paper.
>
> **Response.** Thank you for this comment. We added all relevant citations to the revised paper.

---

> ### Author Response · Authors · 2025-12-03
>
> >Questions:
> >1. The results suggest complex modulation has a dramatic effect on the ring task, but practically no effect on LRGB peptides (table 2). Can the authors explain under what conditions modulation would significantly help?
>
> **Response.**  First, see our response above.
>
> Moreover, we now added a new application of our theory for diagnosing the signal propagation capabilities of trained GNNs. The idea is to plot the signals of various graphs in the dataset at various layers, and see if the next GNN layer changes the location of the signal. Since signals are typically supported across the whole graph, we first multiply the signal by a window, and plug the windowed signal into the next layer, checking if the output has a translated mean location. Here, splitting allows us to see which parts of the signal propagated to which parts of the graph.
>
>
> >2. What is the computational overhead of Schrödinger GNN compared to the baselines, e.g., on LRGB? How much overhead does making the signals epsilon-commuting add in general?
>
> **Response.** We note that exponent is implemented via a Taylor series, i.e., as a polynomial in the Laplacian times $i$. Hence, our method is theoretically as efficient as traditional polynomial filter methods, like ChebNet, GCN, CayleyNet, and ARMANet.
>
> We will add runtime tables to the paper, and a theoretical analysis of the efficiency of exponentiation.
>
>
> >As there were more unitary GNNs recently introduced, could the authors clarify conceptual differences between their method and, e.g., UniConv [6], beyond what is said in appendix A.2?
>
> **Response.** We apologize that this was not clear in our original submission. UniConv [6] is our main baseline, as it is similar to Schrodinger GNN but without modulation. We clarify this in the revised paper, and clearly write UniConv as a line in all of our Tables. We note that Schrodinger GNN outperforms UniConv.

---

### Official Review · Reviewer_CS4h · 2025-10-27

**Soundness:** 3
**Presentation:** 2
**Contribution:** 3
**Rating:** 4
**Confidence:** 4

**Summary:**

This paper uses the notion of observables to model signal propagation. The theoretical contribution is centered on establishing an observable-based model to characterize GNN signal propagation. This model specifically quantifies: (i) signal location, (ii) content concentration around that location, and (iii) propagation flux between graph regions during GNN application. The proposed Schrödinger GNN, based on a unitary graph shift operator and complex modulated signals, theoretically have the potential to achieve directional signal routing.

**Strengths:**

1. This paper introduces the novel concept of observables from quantum mechanics to model and analyze GNN signal propagation, offering a unique and rigorous theoretical framework for GNN.
2. The Schrödinger GNN employs a unitary graph shift operator, which theoretically guarantees norm conservation of the signal, providing a fundamental solution to prevent oversmoothing and information loss.
3. The Schrödinger GNN successfully introduces complex features into the field of graph signal processing, utilizing phase information to encode and control the direction of signal propagation. This opens up a new dimension for GNN feature design, demonstrating that GNNs can be extended from the traditional real space to the richer complex space to encode more complex geometric or dynamic information.

**Weaknesses:**

1. The performance on critical long-range benchmarks is only reported as comparable to state-of-the-art GNNs, failing to demonstrate a significant advantage that would substantiate the theoretical benefit.
2. The authors analyze signal propagation through splitting, but this mechanism is not employed in the actual implementation, which results in a gap between the theory and the practice.
3. There are some citation errors in the draft.

**Questions:**

1. In the introduction, the authors mention oversmoothing and oversquashing. Does the proposed Schrödinger GNN actually address these two issues, or through what mechanism does this model mitigate the limitations of oversmoothing or oversquashing? Since I did not find any experimental results or analyses to substantiate this claim.
2. In the Peptides experiment, the addition of the modulated component to the Schrödinger GNN appears to have little effect on the overall performance. Therefore, is the inclusion of the modulated information truly meaningful in a practical setting? Furthermore, regarding the Unitary GNNs, why such methods not included in the baseline comparison to highlight the advantages of the Schrödinger GNN?
3. Schrödinger GNN introduces complex features. How are these complex features inputted into a traditional GNN model? Furthermore, what is the impact of this conversion operation on the loss or preservation of information?
4. Since Schrödinger GNN involve complex matrix exponentiation operations, please explicitly state its time complexity in the paper.
5. Some errors for missing citations, such as in lines 110, 1228, 1330, and 1341.

---

> ### Author Response · Authors · 2025-12-03
>
> >Weaknesses:
> >1. The performance on critical long-range benchmarks is only reported as comparable to state-of-the-art GNNs, failing to demonstrate a significant advantage that would substantiate the theoretical benefit.
>
> **Response.** Our paper proposes a comprehensive theory for defining and analyzing propagation properties of signals on graphs, when applying GNNs. The theory leads to a novel architecture (Schrodinger GNN). We believe that even being comparable to state-of-the-art, not improving it by a big gap, is enough to merit publication to such a theoretical work. If one adopts this criterion of requiring every paper to significantly outperform existing benchmarks, this would exclude most theoretical advances in deep learning and risk stalling progress in the field. Our work aims to deepen understanding and provide a foundation for future improvements, which we consider an important contribution.
>
> >2. The authors analyze signal propagation through splitting, but this mechanism is not employed in the actual implementation, which results in a gap between the theory and the practice.
>
> **Response.** First, we clearly state that this is a conceptual tool that allows us to interpret how the energy of globally supported signals propagates using our localization analysis. We write in l. 191-195 –
> *”Note that typical signals are not localized about one feature location. For example, the grayscale signal of an image is typically supported across all x, y locations. Hence, the expected location and location variance are not meaningful localization notions for such signals (see Figure 1 for illustration). Still, we can* **conceptually** *apply a localization analysis with observables as follows...”*
>
> To still use the splitting technique in practice, we now added a new application of our theory for diagnosing the signal propagation capabilities of trained GNNs. The idea is to plot the signals of various graphs in the dataset at various layers, and see if the next GNN layer changes the location of the signal. Since signals are typically supported across the whole graph, we first multiply the signal by a window, and plug the windowed signal into the next layer, checking if the output has a translated mean location. Here, splitting allows us to see which parts of the signal propagated to which parts of the graph.
>
>
> >3. There are some citation errors in the draft.
>
> **Response.** Thank you for pointing this out. We fixed these errors in the revised paper.
>
> >Questions:
> >1. In the introduction, the authors mention oversmoothing and oversquashing. Does the proposed Schrödinger GNN actually address these two issues, or through what mechanism does this model mitigate the limitations of oversmoothing or oversquashing? Since I did not find any experimental results or analyses to substantiate this claim.
>
> **Response.** We clearly write that our paper introduces a notion *alternative to oversmoothing and oversquashing*, for depicting the tendency of GNNs to spread the signal across the graph and lose information. This notion is based on observables and their mean and variance. Schrodinger GNNs are designed to address signal propagation in terms of observables, not in terms of oversmoothing and oversquashing.
> Please see, for example,  the abstract:
> *In this paper, we propose an alternative approach for modeling signal propagation, inspired by quantum mechanics, using the notion of observables...*
> and the introduction:
> *We aim to directly study how coherent the signal stays when it is routed between regions of the graph. .For this, we propose in the paper an alternative way to model and probe different aspects of the content of the signal and its flow. Specifically, we model...*
>
>
>
> >2. In the Peptides experiment, the addition of the modulated component to the Schrödinger GNN appears to have little effect on the overall performance. Therefore, is the inclusion of the modulated information truly meaningful in a practical setting? Furthermore, regarding the Unitary GNNs, why such methods not included in the baseline comparison to highlight the advantages of the Schrödinger GNN?
>
> **Response.** Thank you for pointing this out. In the revised paper we added many more experiments and improved the hyperparameter optimization. Now, there are many experiments which show that modulation improves performance. For example TU experimet graph classification ENZYMES (40.3 ± 6.63 without modulation and 43.5 ± 4.89 with) and PROTEINS (69.19 ± 3.01 without modulation and 71.57 ± 2.56 with).
> Please see Tables 2 and 3 in the paper.
>
> Moreover, not adding Unitary GNN was an oversight in the first submission. We now include Unitary GNN in the revised paper, which performs worse than our Schrodinger GNN. See Tables 3 and 4

---

> ### Author Response · Authors · 2025-12-03
>
> >3. Schrödinger GNN introduces complex features. How are these complex features inputted into a traditional GNN model?
>
> **Response.** You can plug complex features into traditional spectral GNNs. The linear filter layers are defined as they are both on real and complex signals. The only thing that you would need to modify is the nonlinearity, which should now be a function defined over the complex plane as its domain.
> However, we introduce a new method - Schrodinger GNN - and we do not need to plug complex features into traditional spectral methods anywhere in our pipeline.
>
>
> >4. Furthermore, what is the impact of this conversion operation on the loss or preservation of information?
>
> **Response.** We would like to note that the whole paper is structured around addressing this question.
>
> See for example, the introduction:
> *Then, we propose a novel spectral GNN, called Schrodinger GNN, which has provably good signal flow properties. Namely, with Schrodinger filters, we can direct the propagation of the signal in any desired direction in the graph.*
>
> *Schrodinger GNNs are based on two main components: a unitary graph shift operator (GSO), and complex modulated signals...  Moreover, Schrodinger GNNs consider some of the input feature channels as encoding an abstract notion of ambient location in the graph. We call these features formal locations. The rest of the feature channels are called the signal. The idea is to be able to shift the signal across the formal location, in any desired direction. ...* **...Moreover, to guarantee that the formal location of signals shifts when applying GNNs, we form in the signal complex oscillations along the direction of each formal location. We show that this leads roughly to a constant speed of the formal location of signals when applying linear Schrodinger filters.**
>
> Then, see section **Achieving Translations via Feature Modulation,** which is devoted to your question.
>
> Read also Section **Improving Signal Routing Through Modulation,** which is devoted to your question.
>
>
>
>
> >5. Since Schrödinger GNN involve complex matrix exponentiation operations, please explicitly state its time complexity in the paper.
>
> **Response.** We note that exponent is implemented via a Taylor series, i.e., as a polynomial in the Laplacian times $i$. Hence, our method is theoretically as efficient as traditional polynomial filter methods, like ChebNet, GCN, CayleyNet, and ARMANet.
>
> We will added a runtime table in the appendix table 8 page 31, and a theoretical analysis of the efficiency of exponentiation in "Computational Complexity" paragraph in page 27 in the appendix.
>
>
> >6. Some errors for missing citations, such as in lines 110, 1228, 1330, and 1341.
>
> **Response.** Thank you for pointing this out. This is now fixed in the revised paper.

---

### Author Response · Authors · 2025-12-03
**Withdrawal of Paper**

Dear Area Chair,
Due to the leak of reviewer identities this year, the standard opportunity to engage in discussion and address reviewer concerns was not given to us. We believe that, under normal circumstances, we could have significantly improved the reviewers’ assessment of our work.
Given this situation, we do not see a viable path for acceptance and have therefore decided to withdraw the paper.

---

### Note · Authors · 2025-12-03

**Comment:**

Due to the leak of reviewer identities this year, the standard opportunity to engage in discussion and address reviewer concerns was not given to us. We believe that, under normal circumstances, we could have significantly improved the reviewers’ assessment of our work. Given this situation, we do not see a viable path for acceptance and have therefore decided to withdraw the paper.

**Withdrawal Confirmation:**

I have read and agree with the venue's withdrawal policy on behalf of myself and my co-authors.